# AutoPartGen:
# Autoregressive 3D Part Generation and Discovery

**Minghao Chen**[1,2]    **Jianyuan Wang**[1,2]    **Roman Shapovalov**[2]    **Tom Monnier**[2]
**Hyunyoung Jung**[2]    **Dilin Wang**[2]    **Rakesh Ranjan**[2]    **Iro Laina**[1]    **Andrea Vedaldi**[1,2]
[1]Visual Geometry Group, University of Oxford    [2]Meta AI
silent-chen.github.io/AutoPartGen

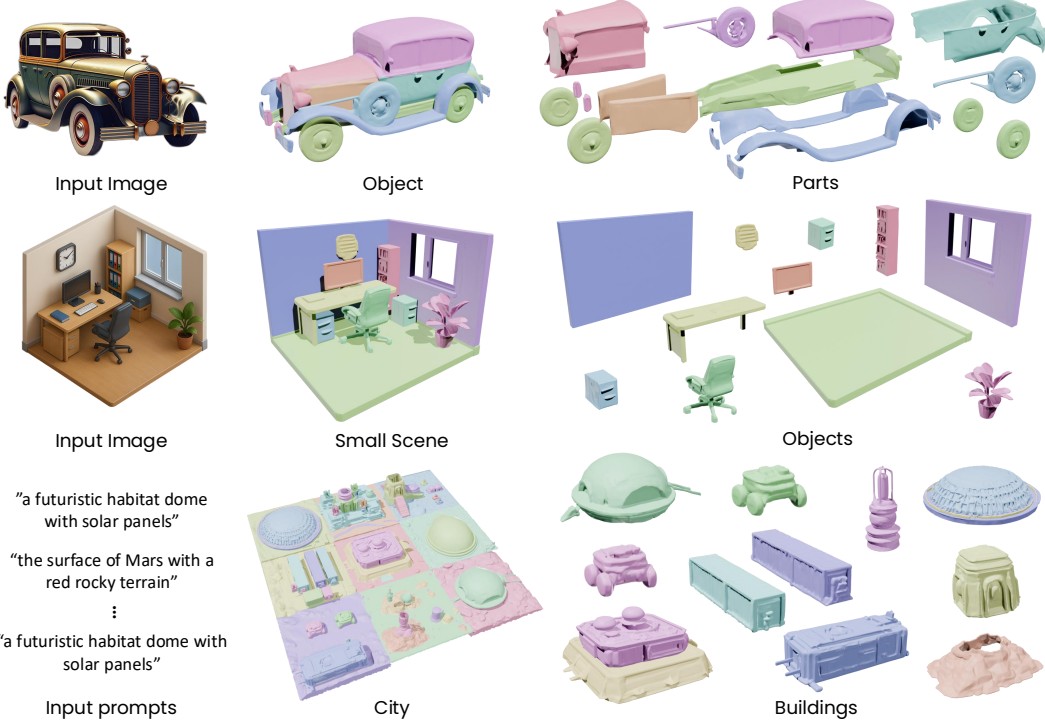

Figure 1: **AutoPartGen** can be *applied*, by itself or in combination with other models, to the generation of compositional 3D objects, scenes and cities starting from 3D models, images or text.

## Abstract

We introduce **AutoPartGen**, a model that generates objects composed of 3D parts in an autoregressive manner. This model can take as input an image of an object, 2D masks of the object's parts, or an existing 3D object, and generate a corresponding compositional 3D reconstruction. Our approach builds upon 3DShape2VecSet, a recent latent 3D representation with powerful geometric expressiveness. We observe that this latent space exhibits strong compositional properties, making it particularly well-suited for part-based generation tasks. Specifically, AutoPartGen generates object parts autoregressively, predicting one part at a time while conditioning on previously generated parts and additional inputs, such as 2D images, masks, or 3D objects. This process continues until the model decides that all parts have been generated, thus determining automatically the type and number of parts. The resulting parts can be seamlessly assembled into coherent objects or scenes without requiring additional optimization. We evaluate both the overall 3D generation capabilities and the part-level generation quality of AutoPartGen, demonstrating that it achieves state-of-the-art performance in 3D part generation.

39th Conference on Neural Information Processing Systems (NeurIPS 2025).

# 1 Introduction

Processing 3D objects, including generating them based on a textual description or an image, is an important aspect of Spatial Intelligence. Current 3D generators often treat objects or even entire scenes as monolithic shells. However, many applications require modeling their compositional structure, decomposing them into well-defined 3D parts to enable reasoning or manipulation at a finer granularity, such as applying textures and materials to each part separately. More specifically, a character in a video game should be decomposable into different parts to support animation or allow the game software to swap clothes or accessories. Windows and doors in the 3D model of a house need to be separate entities to allow user interaction, such as opening or closing them. Similarly, the design of a machine must consist of distinct parts to be functional (e.g., the cogs in a clock) or to enable 3D printing or other kinds of CNC manufacturing.

In this paper, we address the problem of generating 3D objects with a *compositional structure*. We introduce *AutoPartGen*, a new autoregressive model that can *directly* generate a 3D object part by part, building on a powerful latent 3D representation. AutoPartGen is robust, flexible, and scalable. As shown in Fig. 1, AutoPartGen can be applied, either independently or in combination with other models, to generate compositional 3D objects, scenes, or even cities, starting from 3D models, images, or text prompts. AutoPartGen solves three key problems to enable such applications: (i) object-to-parts, where it decomposes an existing 3D object into meaningful parts; (ii) image-to-parts, where the model generates 3D parts from an unstructured input image; and (iii) masks-to-parts, where users can provide 2D part masks to guide the generation. In the first two scenarios, AutoPartGen *automatically* predicts semantically meaningful 3D parts without requiring part annotations. In the third scenario, user-provided masks offer fine-grained control over the model partitioning.

Our autoregressive approach has two key benefits. First, it models the joint distribution over the object parts, ensuring that they fit together cohesively. Second, it enables the model to generate a variable number of parts, which is crucial since the number of parts is not fixed or known a priori.

We build AutoPartGen on recent advances in latent 3D representations and parameterize the 3D surface $x \subset \mathbb{R}^3$ of the object using a latent vector $z \in \mathbb{R}^{M \times d}$, where $M$ is the latent length and $d$ is the latent dimension. We use the 3DShape2VecSet representation [65, 31, 66] and observe, for the first time, that this representation is inherently compositional. Specifically, we show that the concatenation $z = z^{(1)} \oplus z^{(2)} \in \mathbb{R}^{2M \times d}$ of two codes $z^{(1)}$ and $z^{(2)}$ decodes into the union $x = x^{(1)} \cup x^{(2)}$ of the corresponding surfaces $x^{(1)}$ and $x^{(2)}$.

Based on this insight, we propose generating a sequence of latent codes $z^{(1)}, \ldots, z^{(K)}$, each decoding into a corresponding 3D part $x^{(k)}$. Crucially, the generation of each part is conditioned on an overall latent representation of the target 3D object $x$ as well as on all previously generated parts $x^{(1)}, \ldots, x^{(k-1)}$. This conditioning improves the consistency of the generated parts, meaning that they better fit each other compared to the output of the methods that extract parts independently [5, 60].

As noted in [5], object decomposition is an ambiguous problem. For example, a chair might be decomposed into few high-level parts (e.g., seat, back, legs) or a more granular set of components (e.g., individual leg segments, cushion, backrest slats). This choice typically depends on the application or the preferences of the 3D artist creating the asset. We address this ambiguity by making the 3D autoregressive model *stochastic*, using denoising diffusion to generate the next part vector $z^{(k)}$ based on the previously generated parts $z^{(1)}, \ldots, z^{(k-1)}$ and the available evidence (i.e., the full 3D object, an image of the object, or 2D part masks, depending on the application). Importantly, we train a *single* diffusion model capable of handling all three cases.

We evaluate AutoPartGen against state-of-the-art part-aware 3D generators. Compared to the recent PartGen [5], AutoPartGen is easier to implement and maintain (as it does not require training several multi-view image generators) and more accurate. Compared to HoloPart [60], a method that completes a *pre-segmented* outer surface of a mesh to form 3D parts, AutoPartGen is more accurate and significantly more capable, as it can automatically discover parts and reconstruct them from either a 2D image or a "shell" 3D object, optionally guided by 2D masks, not requiring any 3D annotation.

## 2 Related Work

Generating a 3D object from a single image, or even just text, faces an obvious challenge: the 3D object contains significantly more information than the image or the text. This is similar to the problem of generating images or videos from text, and it is solved by learning a prior, or conditional distribution, from billions of data samples. However, data of this size is unavailable for 3D objects. Authors address this problem by involving 2D image or video generators in the 3D generation process. We distinguish two main camps: multi-view direct and single-view latent 3D generation.

**Multi-view 3D Generation.** In multi-view 3D generation, one asks the image generator to do most of the lifting, generating several views of the 3D objects, and thus simplifying extracting a 3D object from them. First, this was done using Score Distillation Sampling (SDS) [42], an idea explored extensively in follow-ups like GET3D [13], ProlificDreamer [54], DreamGaussian [51], Lucid Dreamer [9] which seek to achieve multi-view consistency via iterative (and slow) optimization of a radiance field (NeRF [44] or 3DGS [24]). A significant innovation, pioneered by UpFusion [23], 3DiM [55], Zero-1-to-3 [36] and MVDream [45], was to fine-tune the image generator to directly produce multiple consistent views of the object. By making the image generator more 3D aware, 3D reconstruction becomes simpler, as noted in InstantMesh [57], GRM [58], and others [56, 50].

**Single-view Latent 3D Generation.** The alternative approach is to start from a single image of the object and directly reconstruct the 3D object from it. Because single-view reconstruction is extremely ambiguous, this requires to learn a reconstruction function. This was the path taken, for example, by LRM [19] and others [17, 48]. However, their deterministic reconstruction model cannot cope well with this ambiguity and often produces blurry outputs. Much better results were recently obtained by switching to stochastic 3D reconstruction based on *latent* diffusion. Some of the best single-image 3D reconstructors are based on the 3DShape2VecSet [65] latent representation (also similar to Michelangelo's [67]). Building on it, CLAY [66], DreamCraft3D [47], CraftsMan [30], TripoSG [31], and others [10, 59, 63] are able to generate highly detailed and accurate 3D shapes. We build on this representation as well and show that it also supports compositionality very effectively.

**Composable 3D Generation.** Approaches to composable 3D generation typically start by decomposing objects into constituent parts. One common strategy represents objects as mixtures of primitives, often without semantic labels. For instance, SIF [16] models object occupancy using mixtures of 3D Gaussians. LDIF [15] represents shapes as a set of local deep implicit functions (DIFs), spatially arranged and blended using a template of Gaussian primitives. Methods such as Neural Template [21] and SPAGHETTI [1] achieve decompositions through auto-decoding. SALAD [29] utilizes SPAGHETTI for diffusion-based generation. PartNeRF [52] expands this concept by employing mixtures of NeRFs. NeuForm [32] and DifFacto [39] specifically target part-level controllability. DBW [38] uses textured superquadrics to decompose scenes. In contrast, another research direction emphasizes explicitly semantic parts. PartSLIP [35] and PartSLIP++ [71] segment objects into semantic components from point clouds using vision-language models. Part123 [34] adapts techniques from scene-level approaches like Contrastive Lift [2] to reconstruct object parts. PartSDF [49] learns latent codes for parts using an auto-decoder and then uses SALAD for part prediction. Comboverse [8] leverages single-view inpainting model and 3D generator for composable 3D generation with spatial-aware SDS optimization. Deep Prior Assembly [69] reconstructs 3D scenes from a single image in a zero-shot manner by assembling large models. MIDI [20] extend pre-trained image-to-3D generator to multi-instance generator through costly global attention. CAST [62] reconstructs physically consistent 3D scenes from a single RGB image using occlusion-aware diffusion and GPT-guided physics correction. HoloPart [60], a recent work, starts from the shell of a 3D object and a part-level segmentation for it and performs 3D amodal part completion.

The work most related to ours is PartGen [5]. This squarely sits on the 'multi-view direct' camp (see above). It uses multi-view diffusion models for segmentation and completion of compositional 3D objects from diverse modalities.

**3D Segmentation.** 3D parts can be obtained by segmenting a 3D object (although the resulting parts will generally be incomplete). Some approaches for semantic 3D segmentation such as [68, 28, 53, 25, 2] used neural fields to 'fuse' 2D semantic features in 3D. Contrastive Lift [2] introduced a slow-fast contrastive clustering scheme for 3D instance segmentation. Recent methods such as [26, 64, 43, 3] integrate SAM [27] and often CLIP to model multi-scale concepts, where LangSplat explicitly encodes scale information and N2F2 learns to bind concepts to scales automatically. Neural

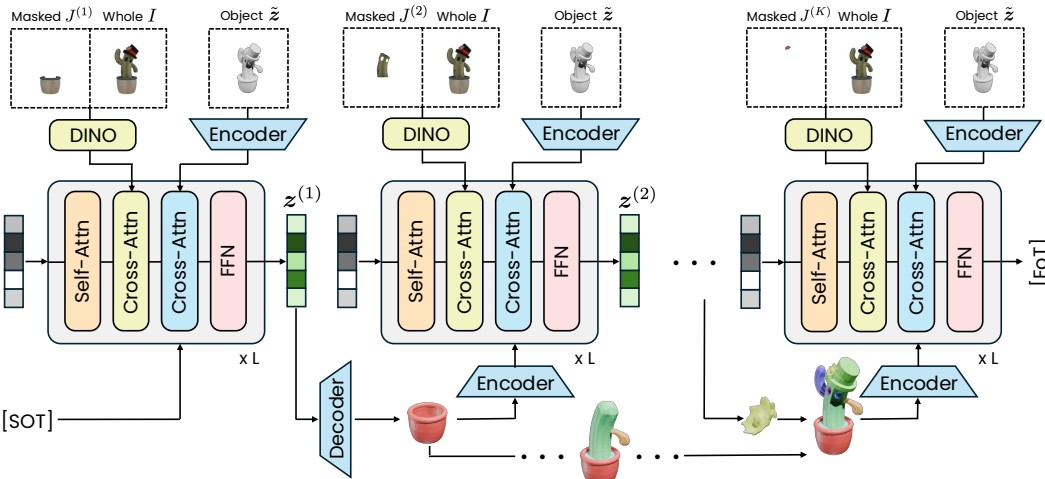

Figure 2: **AutoPartGen generates parts autoregressively.** At each step, a 3D latent diffusion model generate the next part, conditioned on the previously generated parts $z^{(1,...,k)}$, the overall object $\tilde{z}$, and, optionally, an image $I$ of the object and an image $J^{(k)}$ of the part. The latent representation uses 3DShape2VecSet and the diffusion model is a DiT.

Part Priors [4] used learned priors for test-time decomposition. Additionally, efforts to develop 3D 'foundation' models [70, 7] are enabling zero-shot point cloud segmentation across diverse domains.

## 3   Method

Let $x \subset \mathbb{R}^3$ be a *3D object* given by a surface embedded in $\mathbb{R}^3$. We assume that the object is *compositional*, meaning that it can be expressed as the union $x = \bigcup_{k=1}^{K} x^{(k)}$ of $K$ disjoint *parts* $x^{(1)}, \ldots, x^{(K)}$, each of which is also a surface. Concretely, $x$ is usually a 3D mesh created by an artist, and $x^{(k)}$ are the components of the mesh that the artist has manually defined when creating the mesh. These parts are thus defined to facilitate editing of the 3D object or for functional purposes, such as animation. Generally, the same 3D object can have different and equally valid part decompositions.

Our aim is to learn to generate 3D objects $x$ and their part decompositions $x^{(1)}, \ldots, x^{(K)}$. We consider three different scenarios. In the first scenario (object-to-parts), we are given the 3D object $x$, and the goal is to sample a possible decomposition of this object into parts. Furthermore, we may potentially have an object $\hat{x}$ which is incomplete with respect to its constituent parts — a case that may arise if $x$ is acquired by a 3D scanner that cannot look *inside* the object or synthesized by a generator that is not aware of the internal structure of the object, as exemplified by [60]. In this case, therefore, the goal is to also *complete* the parts, thus recovering the complete object $x$ as a byproduct.

In the second scenario (image-to-parts), the starting point is an image $I : \Omega \to \mathbb{R}^3$ of the object, where $\Omega \subset \mathbb{R}^2$ is the (finite) image domain. Inferring the object $x$ from a single image is also known as the *image-to-3D* problem. Here, the task is to also infer its part decomposition $x^{(1)}, \ldots, x^{(K)}$.

In the third scenario (masks-to-parts), we are given, in addition to the image $I$, $K$ 2D part *masks* $M^{(k)} : \Omega \to \{0, 1\}$ that indicate the pixels in the image $I$ that belong to each part $x^{(k)}$. These masks can be defined manually or, more likely, automatically, utilizing a 2D segmentation model. The problem is the same as before, but the 3D parts must match the prescribed masks.

In all these cases, recovering the parts (or the object) is *ambiguous*. These problems are thus *stochastic* and are solved by learning suitable conditional probability distributions: $p(x^{(1)}, \ldots, x^{(K)} \mid x)$ (object-to-parts), $p(x^{(1)}, \ldots, x^{(K)} \mid I)$ (image-to-parts), and $p(x^{(1)}, \ldots, x^{(K)} \mid I, M^{(1)}, \ldots, M^{(K)})$ (masks-to-parts). We develop a single model that can handle all three cases.

## 3.1 Latent 3D shape space

Directly defining, modeling, and learning a distribution on 3D surfaces is difficult. We thus introduce a *latent space*, providing a finite-dimensional parametrization of the surfaces.

We utilize the *VecSet* representation developed by [65]. This representation is based on learning an encoder-decoder pair $(E, D)$. The encoder $E$ takes a collection of $N$ object points $P = \{p_1, \ldots, p_N\} = \mathrm{sample}_N\, \boldsymbol{x}$ and maps them to a latent vector $\boldsymbol{z} = E(P)$. Here $\mathrm{sample}_N$ is a function that samples $N$ random points from the surface of the object $\boldsymbol{x}$, so that $P \subset \boldsymbol{x}$. The decoder takes the latent vector $\boldsymbol{z}$ and a query 3D point $p \in \mathbb{R}^3$ and evaluates the *signed distance function* (SDF) at $p$ as $\mathrm{SDF}(p|\boldsymbol{x}) = D(p|\boldsymbol{z})$. The encoder-decoder pair is thus 'translational', in the sense that it translates one type of representation of the object (the point cloud $P$) into another (the signed distance function $\mathrm{SDF}(\cdot|\boldsymbol{x})$).[1]

In more detail, the encoder $E$ compresses the point cloud $P$ into a sequence $\boldsymbol{z} = (z_1, \ldots, z_M)$ of $M$ tokens $z_i \in \mathbb{R}^D$. The $M \ll N$ tokens are obtained by first subsampling the point cloud $P$ into a much smaller set of points $\tilde{P} = \{p_1, \ldots, p_M\} = \mathrm{sample}_M\, P \subset P$ and then by applying a transformer neural network to the points in $\tilde{P}$ to output $\boldsymbol{z}$. The transformer also attends to the large number of points in $P$ efficiently via cross-attention. The network is designed to be *permutation equivariant*, meaning that the order of the points and tokens is immaterial, explaining the moniker 'VecSet'. The decoder $D$ takes a point $p \in \mathbb{R}^3$ and the tokens $\boldsymbol{z}$ and outputs the value of the signed distance function $\mathrm{SDF}(p|\boldsymbol{x})$, also utilizing a transformer neural network in the form of a Perceiver [22].

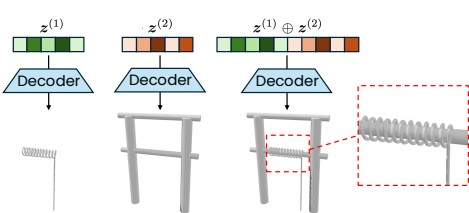

Figure 3: **Compositionality of the VecSet space.** Concatenation of two latents will result in a spatial combined mesh.

The intuition behind this representation is that each token vector $z_i$ encodes a local region of the surface centered at the point $p_i$. However, the transformer allows tokens to communicate globally, which makes this interpretation somewhat loose. Empirically, we have discovered that locality, or at least compositionality, is well supported by the representation. As we show in Fig. 3, the tokens can be concatenated to form a new latent vector $\boldsymbol{z} = \boldsymbol{z}^{(1)} \oplus \boldsymbol{z}^{(2)}$ that decodes into a new surface $\boldsymbol{x} = \boldsymbol{x}^{(1)} \cup \boldsymbol{x}^{(2)}$ that is a good approximation of the union of the two parts, without any retraining.

## 3.2 Latent 3D diffusion

Having established the latent representation $\boldsymbol{z}$ for the shapes, the next task is to learn a model that can sample a shape given some evidence $y$, from a conditional probability distribution $p(\boldsymbol{z} \mid y)$ (for example, $y$ could be the image $I$ of the object). This utilizes (latent) diffusion. In brief, we define a sequence of progressively more noised versions of the latent vector $\boldsymbol{z}$ as $\boldsymbol{z}_t = \sqrt{\alpha_t}\boldsymbol{z} + \sqrt{1 - \alpha_t}\epsilon$, where $\epsilon \sim \mathcal{N}(0, I)$ is a Gaussian noise vector and $\alpha_t$, $t = 0, 1, \ldots, T$ is a schedule of noise levels. Following [46, 14], we introduce the *flow velocity* $\boldsymbol{v}(t, \boldsymbol{z}_t, \epsilon) = (\boldsymbol{z}_t - \sqrt{\alpha_t}\epsilon)/\sqrt{1 - \alpha_t}$. The diffusion model $\hat{\boldsymbol{v}}(t, \boldsymbol{z}_t, \epsilon)$ is trained to predict the flow velocity $\hat{\boldsymbol{v}}(t, \boldsymbol{z}_t \mid y)$ given only the latent vector $\boldsymbol{z}_t$ and the condition $y$, minimizing the loss $\mathcal{L}(\hat{\boldsymbol{v}}) = E_{(y,\boldsymbol{z}),t,\epsilon}\|\hat{\boldsymbol{v}}(t, \boldsymbol{z}_t \mid y) - \boldsymbol{v}(t, \boldsymbol{z}_t, \epsilon)\|^2$ averaged over a training set of evidence-latent vector pairs $(y, \boldsymbol{z})$, a random time step $t$ and noise $\epsilon$.

## 3.3 Autoregressive 3D part generation

The model described in Section 3.1 generates the entire 3D object $\boldsymbol{x}$ (or, more precisely, its latent representation $\boldsymbol{z}$) as a whole. Here, we consider the problem of generating the object parts instead. Our goal is to learn a *single model* that can handle all three part generation scenarios: object-to-part,

---

[1]For this to work, a few technical assumptions are required. We assume $\boldsymbol{x}$ to be the finite disjoint union of closed regular surfaces smoothly embedded in $\mathbb{R}^3$ without self-intersections. This makes the surfaces orientable; then, the surfaces split $\mathbb{R}^3$ into disjoint regions alternating between the outside and inside of the object. In this way, the signed distance function is well defined. The parts $\boldsymbol{x}^{(k)}$ are defined in the same way, and in fact, they are formed by the union of one or more of the closed surfaces that comprise $\boldsymbol{x}$, so that a signed distance function is defined for each part too.

Table 1: **3D part completion.** Reconstruction quality of the parts and whole object. *reproduced.

| Method | 3D Mask | Parts | | | Whole Object | | |
|---|---|---|---|---|---|---|---|
| | | IoU ↑ | F-Score↑ | CD↓ | IoU ↑ | F-Score↑ | CD↓ |
| HoloPart* [60] | ✓ | 0.658 | 0.836 | 0.065 | 0.821 | 0.945 | 0.018 |
| PartGen [5] | ✗ | 0.614 | 0.812 | 0.121 | 0.779 | 0.921 | 0.033 |
| AutoPartGen | ✗ | 0.665 | 0.861 | 0.047 | 0.832 | 0.967 | 0.012 |

image-to-part, and masks-to-part, depending on the inputs $y$ provided. An overview of the pipeline is shown in Figure 2.

To generate an undetermined number of parts $K$, we consider an autoregressive approach, where a single part $\boldsymbol{x}^{(k)}$ is generated each time, based on what was generated before. The model can thus be described as a conditional distribution $p(\boldsymbol{z}^{(k)} \mid y, \boldsymbol{z}^{(1)}, \ldots, \boldsymbol{z}^{(k-1)})$ where $\boldsymbol{z}^{(k)}$ is the latent representation of the $k$-th part and the input $y$ collects the additional evidence available to the model.

The nature of this evidence depends on the reconstruction scenario. In the object-to-part scenario, $y$ is simply the 3D object $\boldsymbol{x}$. In the image-to-part scenario, $y$ is the image $I$ of the object. In the masks-to-part scenario, $y$ is the image $I$ as well as the masked image $J^{(k)} = M^{(k)} \odot I$, denoting which parts should be generated next.

Knowledge of the previously generated parts $\boldsymbol{z}^{(1)}, \ldots, \boldsymbol{z}^{(k-1)}$ is essential as this allows the model to ensure that the next part fits together well with the previously generated ones. Furthermore, in all cases we consider, the evidence $y$ also provides some evidence on the overall shape of the object. As suggested in Fig. 3, we can represent the union of parts by simply concatenating their latent representations. However, for compactness, we found it useful to fuse their codes into one, which we obtain as: $\boldsymbol{z}^{(1,\ldots,k-1)} = E\left(\cup_{j=1}^{k-1} \text{sample}_N D(\cdot \mid \boldsymbol{z}^{(j)})\right)$, where $\text{sample}_N$ is a function that samples $N$ points from the surface of the object defined by the zero level set of the SDF function $D(\cdot|\boldsymbol{z}^{(k)})$, we call this strategy re-encoding.

We found it optional but useful to pin down the overall object by adding to the evidence $y$ a code $\tilde{\boldsymbol{z}}$ for the object as a whole, which is either given outright (object-to-part, $\tilde{\boldsymbol{z}} = E(\text{sample}_N \hat{\boldsymbol{x}})$) or can be obtained by the model itself (image-to-part and masks-to-part, $\tilde{\boldsymbol{z}} \sim p(\boldsymbol{z} \mid I)$) by directly providing an unmasked image to our mode.

With all this, we learn a conditional generator model

$$\boldsymbol{z}^{(k)} \sim p(\boldsymbol{z}^{(k)}|\tilde{\boldsymbol{z}}, \boldsymbol{z}^{(1,\ldots,k-1)}, y), \tag{1}$$

where $y = \phi$ for the object-to-part scenario, $y = I$ for the image-to-part scenario, and $y = (I, M^{(k)})$ for the masks-to-part scenario. The generation process stops when all the input masks have been processed, if available, or when the model outputs a predefined special [EoT] token, representing empty shape. In practice, we represent the [EoT] token using latents whose values are all zeros.

Based on Section 3.2, learning the distribution Eq. (1) amounts to learning a velocity field $\hat{\boldsymbol{v}}(t, \boldsymbol{z}_t \mid \tilde{\boldsymbol{z}}, \boldsymbol{z}^{(1,\ldots,k-1)}, y)$. During inference, we use *classifier-free guidance* (CFG) [18] to modulate the strength of the conditioning. In the most general case, the model is conditioned by the overall (partial) object $\tilde{\boldsymbol{z}}$, the previously generated parts $\boldsymbol{z}^{(1,\ldots,k-1)}$, and a masked image pair $y = (I, J^{(k)})$. We modulate the importance of the geometric and visual conditioning as follows:

$$\hat{\boldsymbol{v}}_{\text{CFG}}(t, \boldsymbol{z}_t \mid \tilde{\boldsymbol{z}}, \boldsymbol{z}^{(1,\ldots,k-1)}, y) = w_{\text{img}} \left( \hat{\boldsymbol{v}}(t, \boldsymbol{z}_t \mid \tilde{\boldsymbol{z}}, \boldsymbol{z}^{(1,\ldots,k-1)}, I, J^{(k)}) - \hat{\boldsymbol{v}}(t, \boldsymbol{z}_t \mid \tilde{\boldsymbol{z}}, \boldsymbol{z}^{(1,\ldots,k-1)}) \right)$$
$$+ w_{\text{geom}} \left( \hat{\boldsymbol{v}}(t, \boldsymbol{z}_t \mid \tilde{\boldsymbol{z}}, \boldsymbol{z}^{(1,\ldots,k-1)}) - \hat{\boldsymbol{v}}(t, \boldsymbol{z}_t, \emptyset) \right) + \hat{\boldsymbol{v}}(t, \boldsymbol{z}_t, \emptyset), \tag{2}$$

where $w_{\text{img}}$ and $w_{\text{geom}}$ modulate, respectively, image and geometry conditioning. The different inputs are implemented by first encoding into tokens, which are then cross-attended by a transformer neural network computing the flow velocity. Hence, to suppress an input we simply replace it with dummy tokens. In the same way, we randomly drop some input at training time to allow the model to learn to use any required subset of the inputs.

**Discussion** Here, we contrast our model to prior works and justify its design. The most straightforward approach to part generation is to sample each part $\boldsymbol{x}^{(k)}$ independently from a 'marginal'

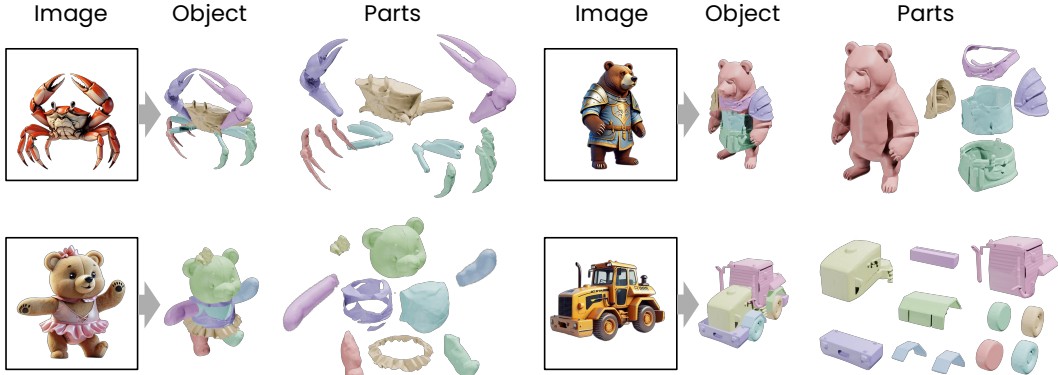

Figure 4: **Image-to-parts scenario.** Given an input image, AutoPartGen recovers a compositional 3D object made up of several meaningful and complete parts.

distribution $p(\boldsymbol{x}^{(k)})$. However, this model lacks a mechanism to tie the parts together and would result in a soup of random, uncoordinated parts. The simplest such mechanism is to provide evidence $y$ for the overall shape of the 3D object. For instance, in the image-to-3D case, $y = I$ could be a 2D image of the object, and we may sample parts from the conditional distribution $p(\boldsymbol{x}^{(k)} \mid I)$. While $I$ constrains the shape and position of the possible parts, these are still quite ambiguous. This explains why PartGen [5] conditions part generation on a multi-view image $y = I_{\mathrm{MV}}$ of the 3D object $\boldsymbol{x}$, and HoloPart [60] starts from a (partial) 3D reconstruction $y = \hat{\boldsymbol{x}}$ of the object itself.

Even then, the reconstruction context $y$ is likely insufficient because there is no indication of *which* part should be reconstructed next. We could sample the parts in a random order, but this would not be very efficient. Furthermore, because the part decomposition is not unique, we would still need to extract a coherent subset of parts from the 'part soup' so obtained.

Prior works address this issue by explicitly telling the 3D reconstruction model which part to extract next. PartGen does so by providing a multi-view image $J_{\mathrm{MV}}$ of the part, and HoloPart by providing a 3D mask $M_{\mathrm{3D}}$ of the part, defining distributions $p(\boldsymbol{x}^{(k)} \mid I_{\mathrm{MV}}, J_{\mathrm{MV}})$ and $p(\boldsymbol{x}^{(k)} \mid \hat{\boldsymbol{x}}, M_{\mathrm{3D}})$, respectively. Hence, the problem of generating a coherent collection of parts is offloaded to a mechanism external to the 3D reconstructor. On the contrary, our 3D generator/reconstructor makes this determination by itself, operating autoregressively, one part at a time, without additional models.

## 4 Experiments

We first give the implementation details of AutoPartGen, including network architectures, training procedures, and datasets. We then demonstrate its performance under various conditions, highlighting its versatility for different applications. Next, we compare our approach with state-of-the-art 3D completion methods and provide ablation studies to analyze key design choices. Finally, we showcase several applications of AutoPartGen.

### 4.1 Implementation Details

**Architecture.** Our architecture builds upon the 3DShape2VecSet [65] framework, with some modifications. Specifically, we increase the input points of the VAE encoder to 32K, and utilize both point coordinates and normals as input features to better capture fine-grained geometric details. The diffusion model is implemented as a DiT [41] with a width of 2048 and 24 layers. For image-conditioned generation, we use DINOv2 [40] to encode the input image $I$ and part-masked images $J^{(k)} = I \odot M^{(k)}$ independently. The resulting features are concatenated along the channel dimension and passed through a small MLP to match the diffusion transformer input. We provide more details in the supplementary material.

**Training.** We use the AdamW optimizer with a learning rate of 1e-4 and train the model for 500K iterations on 256 NVIDIA H100 GPUs. Training the full model takes approximately 4 days. More details on hyperparameters and data preprocessing are provided in the supplementary material. During

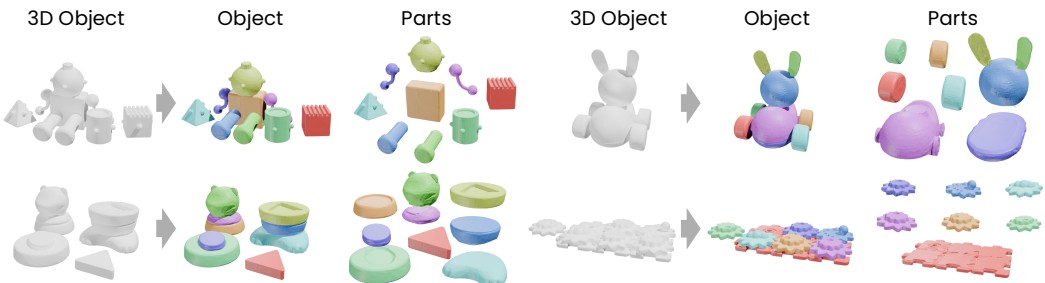

Figure 5: **Object-to-parts scenario.** Given an input 3D object, AutoPartGen regenerates it as a composition of meaningful and complete 3D parts.

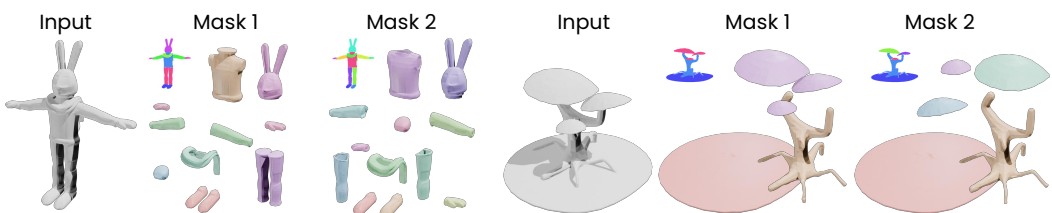

Figure 6: **Masks-to-parts scenario**. AutoPartGen reconstructs a compositional 3D object guided by user-provided 2D part masks. Varying these masks yields different decompositions, potentially at different levels of granularity.

training, we randomly drop the image condition, the geometry condition, or both with probabilities of 0.05 each. For CFG, we use $w_{\text{img}} = 7$ and $w_{\text{geom}} = 4$ as the default setting.

**Training Data.** Our training data pipeline draws inspiration from PartGen [5], but is substantially scaled to encompass approximately 300K assets and 2M individual parts. We start with a collection of 1.8M 3D assets, all licensed that permit AI training. Each asset is stored in glTF/GLB formats, which contains multiple meshes in it and embeds a hierarchical structure. To manage complexity, if an asset contains more than a predefined maximum of 15 meshes, we iteratively merge meshes from the bottom up, until the mesh count is within this limit. To prepare for training VAE and diffusion models, we compute a truncated signed distance for each part in a normalized space and also render different views for image-conditioned cases. More details are included in the supplementary materials.

### 4.2 Object, Image and Masks to 3D Parts Generation

We test AutoPartGen with different types of inputs to demonstrate its versatility. Specifically, we consider: (1) image-to-parts generation from a single input image, where the images are generated by text-to-image (2D) generators; (2) object-to-parts decomposition from a full 3D mesh, with meshes sourced from Google Scanned Objects [11]; and (3) masks-to-parts generation with user-provided 2D part masks, where the masks are taken from PartObjaverse-Tiny [61]. Figures 4 to 6 qualitatively demonstrates that AutoPartGen produces accurate and consistent 3D parts across all these input types.

### 4.3 Comparison to the State-of-the-Art

**Evaluation Protocol.** We use PartObjaverse-Tiny [61] for evaluation, filtering out very small parts following the protocol of [60]. This dataset comprises objects from diverse categories with manually annotated 3D part segmentations. We use standard metrics to assess the quality of the reconstructed geometry: Intersection-over-Union (IoU), Chamfer Distance (CD), and F-score. IoU is calculated on $64^3$ voxel grids, and the F-score adopts a distance threshold of $0.02$. We report the quality of the reconstruction of individual parts and of the overall object after merging them.

**Results.** We compare AutoPartGen to two recent methods: PartGen [5] and HoloPart [60]. We focus on mask-controlled part generation. Recall that HoloPart takes as input the overall partial 3D object and 3D part segmentations and outputs the complete parts. We adapt both PartGen and AutoPartGen

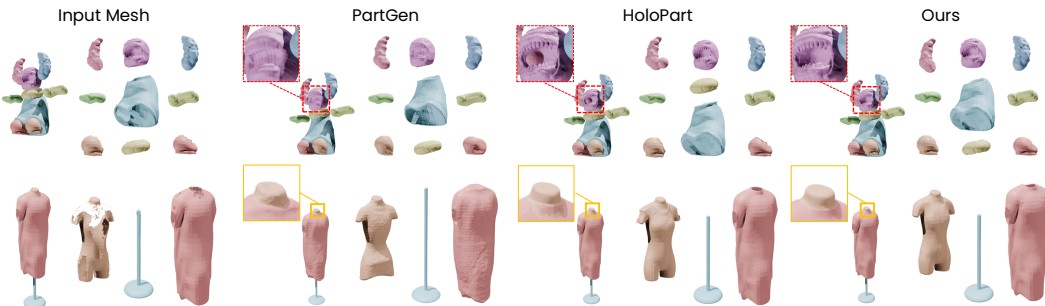

Figure 7: **Visual comparison of different completion methods.** Our approach achieves better geometric coherence by considering previously generated parts in context.

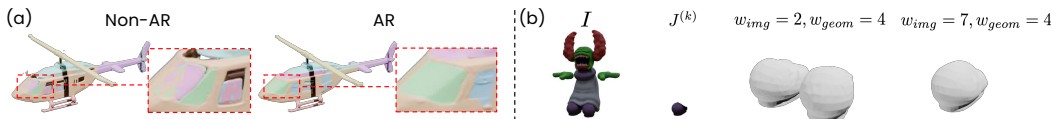

Figure 8: **Ablation Study.** (a) Without autoregressive generation, parts overlap and intersect. (b) Increasing image guidance $w_{\text{img}}$ encourages the model to follow image mask, while a larger geometry guidance $w_{\text{geom}}$ biases the generation towards a generic part distribution and order in the data.

to solve the same problem. For AutoPartGen, we provide the overall partial 3D mesh and 2D part masks (a variant of masks-to-parts). For PartGen, we supply four masked views of each part, which are compatible with its input requirements.

The results in Table 1 show that HoloPart outperforms PartGen, likely due to its access to more comprehensive input information (the partial 3D object and 3D part masks). Nevertheless, AutoPartGen surpasses both baselines across all key metrics: IoU, F-score, and Chamfer Distance. This advantage holds true for both part completion and overall object reconstruction, indicating that AutoPartGen generates geometrically precise parts that form a well-formed and coherent whole.

## 5    Ablation Study

We ablate three factors in AutoPartGen: the autoregressive design, the guidance scale, and the re-encoding strategy. We provide qualitative evidence for the first two factors and report quantitative results for all three.

**Autoregressive modeling.** We evaluate the contribution of the autoregressive design by removing the autoregressive condition $z^{(1,...,k-1)}$ in the masks-to-part setting. This is the only setting where removal is possible, since external part guidance specifies which part to generate next and when to stop. As shown in Figure 8(a), removing this conditioning causes parts to overlap and intersect. Quantitatively, Table 2 shows that enabling autoregression improves IoU and F-score, and reduces Chamfer Distance (CD), indicating better geometric fidelity and coherence.

**Effectiveness of guidance.** We analyze the impact of *image guidance* $w_{\text{img}}$ and *geometry guidance* $w_{\text{geom}}$ in masks-to-parts generation. Because image and geometry conditions are randomly dropped during training, the model learns a data-driven prior for part partitioning and ordering when no image condition is present. As shown in Figure 8(b), increasing $w_{\text{img}}$ aligns parts more closely with input image masks, while a higher $w_{\text{geom}}$ biases generation toward the learned prior of plausible part structures. Quantitatively, Table 3 shows that moderate guidance values (for example, $w_{\text{geom}}=4$ and $w_{\text{img}}=7$) strike the best balance, achieving peak performance in IoU, F-score, and Chamfer Distance.

**Effect of re-encoding.** We further compare three ways to aggregate the VecSet tokens of different parts: (i) *Re-encoding*, which first decodes different parts into meshes, concatenates them, and re-encodes the concatenated mesh into a fixed-length latent; (ii) *Concat*, which directly feeds the concatenated latents into the cross-attention layer of the diffusion model; and (iii) *Latent fuser*, use a 6-layer Perceiver-style module that fuses the tokens into a fix length of 512 latent tokens. All models

Table 2: **Ablation study on autoregressive.** Autoregressive generation clearly show better results in terms of the part completion and overall object coherence. The models are only trained for 200 epochs. Here CD denotes Chamfer Distance.

| Autoregressive | Part Completion | | | Overall | | |
|:---:|:---:|:---:|:---:|:---:|:---:|:---:|
| | IoU ↑ | F-Score ↑ | CD ↓ | IoU ↑ | F-Score ↑ | CD ↓ |
| ✗ | 0.574 | 0.795 | 0.067 | 0.783 | 0.917 | 0.031 |
| ✓ | 0.633 | 0.825 | 0.052 | 0.811 | 0.934 | 0.022 |

Table 3: **Effects of different guidance scales**. We report the three metrics on the part completion task, where CD denotes the Chamfer Distance.

| Geometry/Image | 5 | | | 7 | | | 9 | | |
|:---:|:---:|:---:|:---:|:---:|:---:|:---:|:---:|:---:|:---:|
| | IoU | F-score | CD | IoU | F-score | CD | IoU | F-score | CD |
| **2.5** | 0.650 | 0.847 | 0.051 | 0.639 | 0.839 | 0.053 | 0.632 | 0.833 | 0.052 |
| **4** | 0.657 | 0.854 | 0.050 | 0.665 | 0.861 | 0.047 | 0.648 | 0.852 | 0.052 |
| **5** | 0.635 | 0.841 | 0.062 | 0.662 | 0.857 | 0.049 | 0.647 | 0.851 | 0.051 |

Table 4: **Effect of re-encoding**. All models are trained for 150 epochs with 512 tokens per part under the same setup.

| Method | IoU ↑ | F-Score ↑ | Chamfer Distance ↓ |
|:---|:---:|:---:|:---:|
| Re-encoding | 0.627 | 0.815 | 0.055 |
| Concat | 0.611 | 0.804 | 0.059 |
| Latent Fuser | 0.608 | 0.802 | 0.061 |

are trained for 150 epochs with 512 tokens per part under the same setup. As summarized in Table 4, all three methods perform similarly, with re-encoding slightly outperforms the two other strategies in terms of IoU, F-score, and CD.

## 5.1 Applications

**3D Scene Generation.** Our method's capability for decomposable object generation naturally extends to entire 3D scenes. As illustrated in Figure 1 (middle row), given an isometric view of a small scene, our approach automatically generates individual scene objects such as chairs, clocks, plants, and tables in a decomposable manner. This decomposable nature facilitates flexible manipulation and editing of individual scene components.

**City Generation.** Figure 1 (bottom row) further showcases our method's potential for large-scale outdoor scene generation. Drawing inspiration from SynCity [12], we employ a text-to-image genera-tor to produce diverse tile images from text prompts. These tiles are subsequently assembled to form coherent cityscapes, enabling scalable and controllable generation of complex urban environments. Further examples and qualitative demonstrations are provided in the supplementary material.

## 6 Conclusion

We introduced AutoPartGen, an autoregressive model for compositional 3D part generation. By leveraging latent 3D representations, our method generates coherent object parts sequentially. The same model can handle different input types, including images, 2D masks, and 3D meshes. Au-toPartGen outperforms existing methods in part completion and coherence while simplifying the overall pipeline. Our experiments demonstrate its effectiveness across various tasks and applications, highlighting its potential for scalable and controllable 3D content creation.

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

# Supplementary Material

This supplementary material provides additional details and results to complement the main paper. It includes the following sections:

- **Implementation Details:** Detailed descriptions of model architectures, training and inference procedures, and evaluation protocols.
- **Additional Comparisons with PartGen:** Extended evaluation against PartGen.
- **3D Scene Generation:** Additional qualitative examples for scene generation.
- **City Generation:** Details of the city generation pipeline and additional visual results.
- **Failure Case:** Visualization of failure cases in AutoPartGen.
- **Limitations and Broader Impact:** Discussion of limitations and potential societal implications of AutoPartGen.

## A    Implementation Details

### A.1    Training

Our model consists of two primary components: a 3D Variational Autoencoder (VAE) and a diffusion model.

**3D Variational Autoencoder.** We adopt the 3D representation from 3DShape2VecSet, extending it to a larger model capacity compared to the original VAE [65]. Our VAE architecture comprises an 8-layer encoder with a dimension of 768 and a 16-layer decoder with a dimension of 1024. The model is trained on approximately 1.7M 3D assets, with data augmentation techniques including point cloud rotations, as suggested by Dora [6]. We employ a signed distance function (SDF) representation for smoother isosurface extraction. During training, we supervise the VAE using a combination of surface normal loss, Eikonal loss, and KL divergence regularization, weighted by 10, 0.1, and 0.001, respectively, following TripoSG [31]. To learn a single signed distance field, we calculate a combination of L1 and MSE loss on a total of 24,576 points per shape: 8192 each from surface points, near-surface points, and randomly in the volume. We randomly vary the number of input tokens between {512, 2048} during training. The model is optimized using AdamW with a learning rate of $1e-4$, linearly warmed up from $1e-5$ over the first 3 epochs. We use a batch size of 1536 and set the weight decay to 0.01. Training is conducted on 128 NVIDIA H100 GPUs for 150 epochs.

**Pretraining and fine-tuning.** For diffusion training, we first pretrain a general image-to-3D model on the same 1.7M assets. The diffusion backbone follows DiT [41], with 24 transformer layers and hidden dimension 2048. We train with a fixed token length of 512 for 300 epochs, learning rate $1e-4$, and batch size 10 per GPU on 128 GPUs. We then fine-tune the model to additionally condition on masked image and geometry tokens on the part dataset in an autoregressive manner for approximately 300k steps. The image condition is encoded with DINO-v2 [40], and the geometry token is encoded with our trained 3D VAE. To reduce computation, geometry tokens are used only in the first 12 transformer layers. We apply condition dropping with probability 0.05 independently for the different combination of inputs simultaneously. Fine-tuning uses AdamW with weight decay 0.01 and batch size 6 per GPU on 128 GPUs. Subsequently, we increase the token length to 2048 and continue training for an additional 100k steps on 256 GPUs with batch size 1 per GPU. We adopt the DDIM scheduler [46] with 1000 steps, use v-prediction, and a zero signal-to-noise ratio [33].

**Ordering of parts.** In the *masks-to-parts* setting, the order of the input masks specifies the order in which parts are generated. Both unmasked and masked images are provided so the model can learn spatial relationships between each part and the whole object in 3D space. In practice, the model can accurately infer very small parts (less than $0.1\%$ of the object volume) from very small masks, and at inference time, users can interactively adjust the image guidance scale to control how strongly masks and images influence generation. In the *automatic setting*, a predefined order is used during training and followed at inference. Training assets are defined in a canonical space, and parts are sorted lexicographically by their axis-aligned bounding boxes: bottom to top (Z), then left to right (X), then front to back (Y), following Blender's ZXY axis convention. Concretely, the minimum Z

values are compared first; if they are similar, the minimum X values are compared, and if still similar, the minimum Y values are used..

## A.2 Inference

In all scenarios, we use 50 denoising steps during inference. For the object-to-parts setting, geometry guidance is set to 10, while image guidance is disabled (set to 0). For both the image-to-parts and masks-to-parts settings, we first perform image-to-3D reconstruction to obtain the overall object shape. When user-provided masks are available, we apply the default guidance setting, with image guidance set to 7 and geometry guidance set to 4.

## A.3 Evaluation

The most relevant baseline to AutoPartGen is PartGen. We compare the two methods under the object-to-parts setting, without incorporating any user inputs. For this comparison, we use objects from the Google Scanned Objects (GSO) dataset [11].

When users provide masks to guide part partitioning, we compare our method with recent approaches, including HoloPart and PartGen. For evaluation, we use the PartObjaverse-Tiny dataset [61]. To exclude negligible parts, we filter out objects containing segments that occupy only a small fraction of the total object volume as in [60].

## B    Additional Comparison with PartGen

To further highlight the improvements over PartGen, we provide a qualitative comparison in Figure 9. As shown in the figure, AutoPartGen produces sharper and more detailed meshes, as highlighted by the red circle. Additionally, the autoregressive generation in AutoPartGen avoids over-generation issues seen in PartGen, which arise from its lack of explicit modeling of the joint distribution of different parts. This is a key capability addressed by AutoPartGen.

## C    3D Scene Generation

Small scene generation is a natural extension of our method. Specifically, we begin by providing prompts such as "an isometric view of an office" or "an isometric view of a small bedroom" to an off-the-shelf 2D text-to-image generator to produce corresponding images. We first generate the overall shape of the small scene. Subsequently, we apply AutoPartGen again to decompose the scene into distinct components such as "chair", "table" and so on. More qualitative examples are provided in Figure 10.

## D    City Generation

As shown in  Figure 1 of the main paper, we demonstrate the ability of AutoPartGen to generate 3D cities by integrating it into the SynCity [37] pipeline, replacing its original 3D generator. Specifically, the pipeline begins with text prompts generated by a large language model, which are then used to guide a 2D image generator in creating isometric views of individual tiles, taking into account the context of neighboring tiles. These generated images are then passed to AutoPartGen to produce compositional 3D tiles.

We present additional examples in Figure 11. As illustrated, AutoPartGen can be seamlessly integrated into the city generation pipeline, generating different cities, such as a 'medieval town' or a "solarpunk city". For further details on prompt generation and 2D image synthesis, please refer to the SynCity paper.

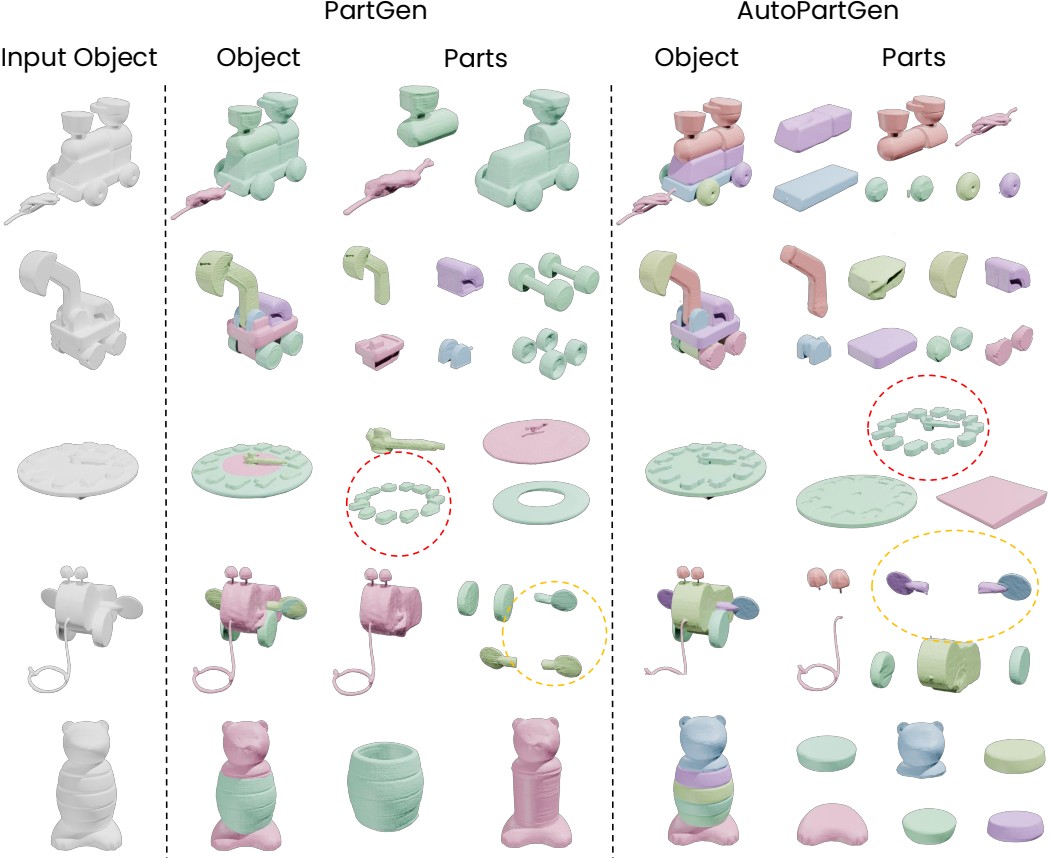

Figure 9: **Comparison between AutoPartGen and PartGen.** AutoPartGen produces more accurate geometry, as highlighted in the red circle. Additionally, its autoregressive generation prevents the overlapping parts observed in PartGen, as shown in the yellow circle.

# E  Failure Cases

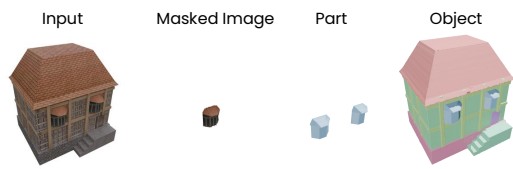

Figure 12: **Failure Case**. When there are identical parts, the model sometimes will try to predict the parts together even if masks are given.

We present a failure case in Figure 12. As shown in the figure, when the object contains identical parts, the model attempts to predict multiple parts together, even when only one is masked. We conjecture that this behavior comes from the bias in the training data, where some artists may have grouped identical parts into a single mesh. Additionally, as discussed in the ablation study, users may adjust the guidance scale to enforce stronger adherence to the image, which can amplify this effect. Due to the model's autoregressive nature, each prediction depends on previously generated parts. In the example shown in Figure 12, where both window instances are generated together, when the mask for the second window is given as input, the model's prediction will be (nearly) empty, since the model has already generated that content. Hence, despite this potential failure mode, the model produces coherent results, maintaining consistency in the overall shape.

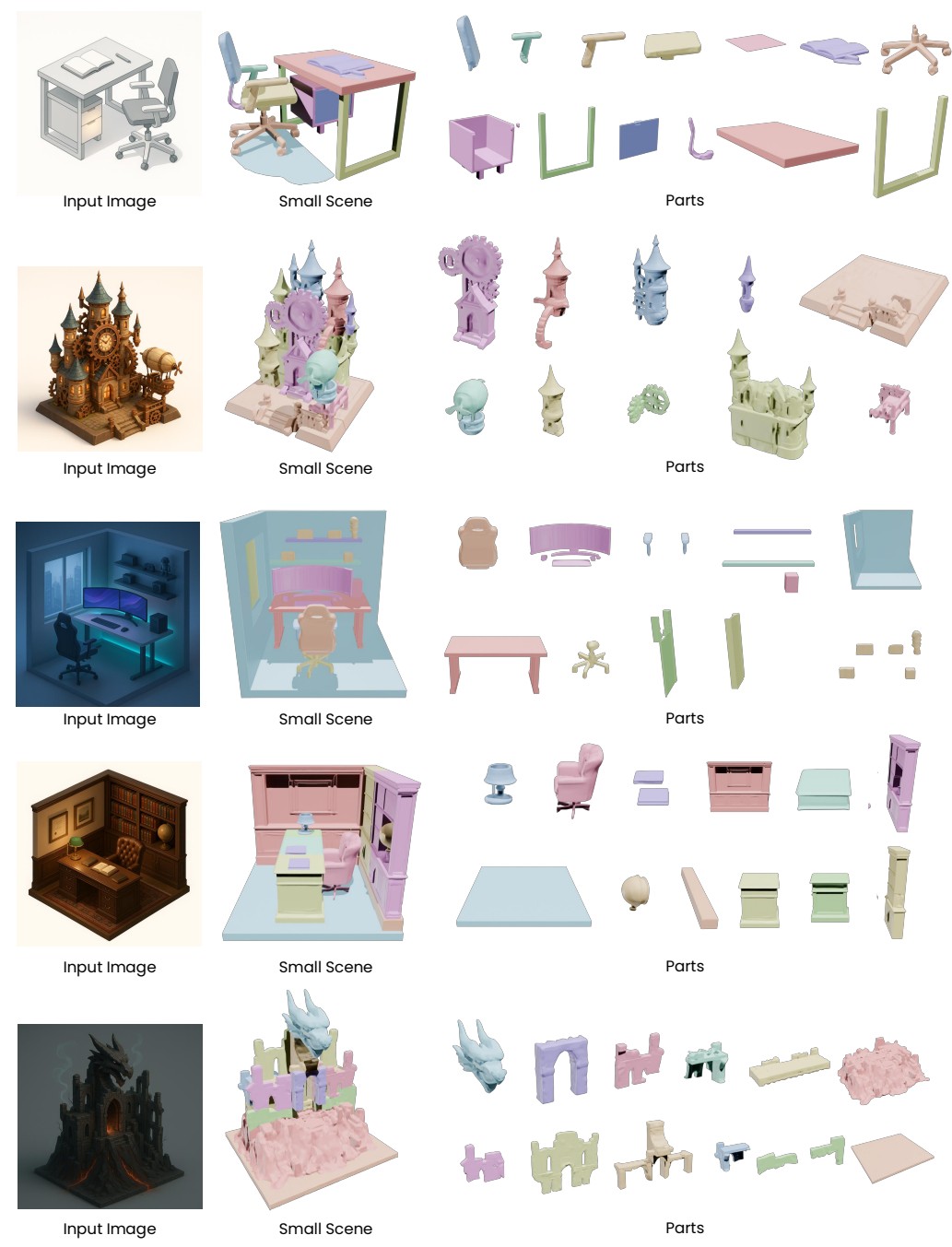

Figure 10: **3D scene generation.** AutoPartGen generates 3D scenes while decomposing them into their constituent elements. The input images are generated by a 2D text-to-image generator.

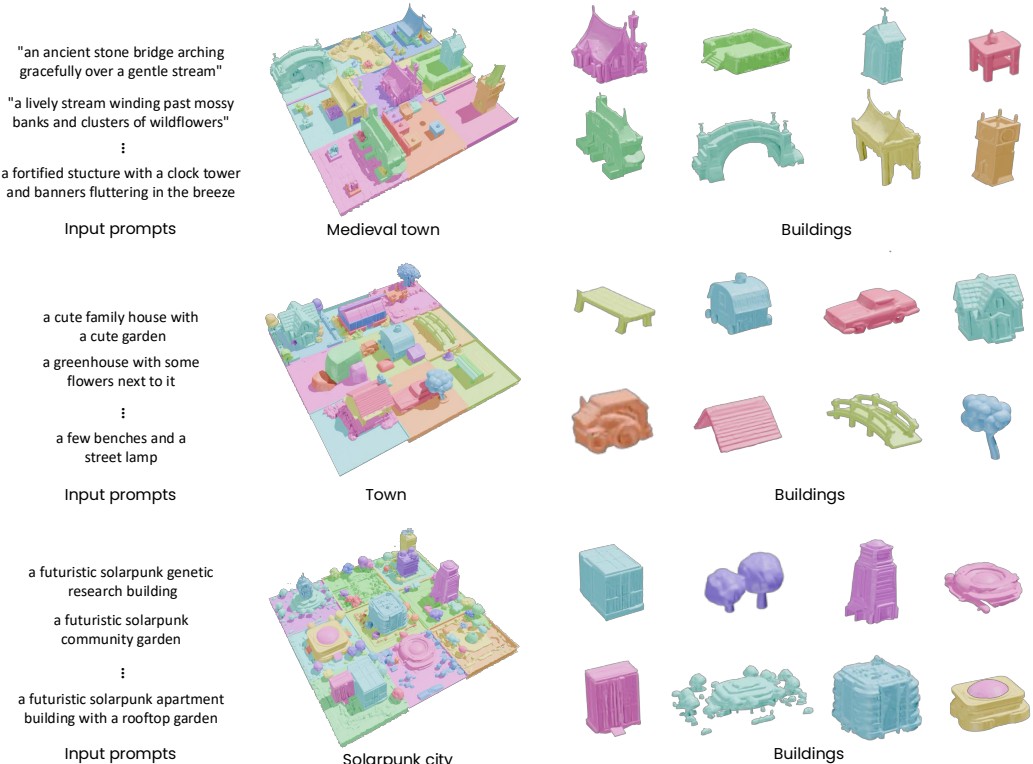

Figure 11: **City Generation.** We showcase AutoPartGen on larger scenes by integrating it within Syncity [12]. From top to bottom, the images depict a medieval town, a cozy town, and a solarpunk city.

## F  Limitations and Boarder Impact

**Limitations.**  While AutoPartGen demonstrates strong performance across all three scenarios, it also has some limitations that point to potential directions for future improvement. First, the current model can only generate bounded scenes, as it inherits the spatial constraints from the underlying VAE latent space. Extending the framework to support unbounded world generation, where scenes can grow or evolve without a predefined spatial limit, would be both an interesting challenge and a promising research direction. Second, the method currently lacks explicit control over the granularity of part partitioning, except in the masks-to-parts setting, where masks can be provided as a way of control. In the image-to-parts and object-to-parts settings, the decomposition of parts can vary. In future iterations, one may incorporate high-level controls for granularity levels, such as 'simple', 'medium', and 'complicated', to make the system more flexible and interactive. Finally, the model learns part distributions directly from the training data, which introduces the risk of bias being inherited from the dataset.

**Broader Impact.**  Although our model is trained on a large amount of data, it may still exhibit biases that reflect imbalances in the underlying distribution. These biases can influence downstream tasks and should be carefully evaluated before practical deployment. To mitigate potential misuse or harmful applications, we recommend implementing safeguards and conducting thorough audits to ensure responsible usage. Additionally, the training process requires substantial computational resources, particularly in terms of GPU usage. This raises concerns about energy consumption and the associated environmental impact, which should be taken into account when scaling or deploying the model in real-world settings.

