# OpenReview forum: "AutoPartGen: Autoregressive 3D Part Generation and Discovery"
_NeurIPS.cc/2025/Conference — NeurIPS 2025 poster_

### Official Review · Reviewer_Hib7 · 2025-06-03

**Clarity:** 4
**Significance:** 3
**Originality:** 4
**Rating:** 4
**Confidence:** 3

**Summary:**

The method generates 3D objects by dividing them into smaller parts and synthesizing each part independently. It supports generation from either an image or a 3D object, with optional part-specific masks. The process follows an autoregressive strategy, where each part is conditioned on all previously generated parts.

**Questions:**

1. Generation Order: How is the order of part generation determined? Can we predict which parts are generated first? During training, how do you decide the sequence of parts?
2. Canonical Positioning: Is the canonical position of each part defined based on the final assembled object, or can the object shape and part positions be learned independently?
3. Detail Level of Parts: How can we assess the level of detail in generated parts? For example, in the room scene (Fig. 1 middle), the “chair” part is generated as a whole, but generating a chair object as parts typically involves distinct components like handles and wheels.
4. Necessity of Part-wise Generation: If the model is capable of generating full objects, what is the motivation for generating parts separately?

**Ethical Concerns:**

["NO or VERY MINOR ethics concerns only"]

**Final Justification:**

i see no reason to change my rating

**Limitations:**

1. The method cannot handle objects without clear parts, such as natural elements like seas or rivers, or highly complex objects like trees with many small leaves.
2. The paper does not address texture generation, which limits the realism and completeness of the generated 3D parts.

**Quality:**

3

**Strengths And Weaknesses:**

Strengths:
1. New approach for generating 3D assets, by autoregressive generating parts of the object. This approach can provide more information about the generated object about it's decomposition.
2. Three tasks involving part generation. image-to-parts, object-to-parts, masks-to-parts.

Weaknesses:
1. The method cannot handle objects without clear parts, such as natural elements like seas or rivers, or highly complex objects like trees with many small leaves.
2. Applying the same resolution to all parts may result in some parts being overly detailed, while others appear low-resolution, leading to inconsistent quality.
3. The paper does not address texture generation, which limits the realism and completeness of the generated 3D parts.

---

> ### Author Rebuttal · Authors · 2025-07-29
>
> > The method cannot handle objects without clear parts, such as natural elements like seas or rivers, or highly complex objects like trees with many small leaves.
>
> Thanks! It is not clear how one would decompose *stuff* categories like "water" into parts, so we think it is legitimate to leave this out of scope. As for the object complexity, the main problem is one of *scalability* rather than of principle. Given a tree, AutoPartGen can extract the canopy, trunk, roots, and branches, but it is not trained to extract thousands of small leaves. However, recent high-resolution 3D generators [*1] suggest that this might be possible in the future.
>
> > Applying the same resolution to all parts may result in some parts being overly detailed, while others appear low-resolution, leading to inconsistent quality.
>
> This is a good point, and allocating a variable number of tokens to different parts is certainly an interesting direction for future work. In our current setup, we follow the original 3DShape2VecSet [59] design and adopt a fixed-length VecSet representation for all parts. As in the original paper [59], the object is encoded into a fixed-length latent regardless of the object complexity. This approach is simple and robust, as demonstrated in prior works [29, 55, 59, 60].
>
> > The paper does not address texture generation, which limits the realism and completeness of the generated 3D parts.
>
> Similar to VecSet [59] and other related works [28, 29, 55, 60, 61], our model focuses solely on generating 3D geometry without textures. The VecSet representation, which we build on, does not include appearance information. However, VecSet is typically used in conjunction with texture generation models such as TextureGen [*2] or MV-Adapter [*3], as demonstrated in [28, 29, 60], to generate appearance after geometry is established. We plan to further explore this direction in future work, aiming to jointly model both geometry and appearance.
>
>
> > Generation Order: How is the order of part generation determined? Can we predict which parts are generated first? During training, how do you decide the sequence of parts?
>
> Thank you for your helpful suggestion. In the masks-to-parts scenario, the order of the input masks determines the order in which parts are generated. In the automatic setting, we use a predefined order during training, which the model also follows during inference. Specifically, the training assets are defined in a canonical space, and we sort the parts based on their bounding boxes in the order of bottom to top (Z), left to right (X), and front to back (Y), following Blender’s ZXY axis convention. We first compare the minimum Z values of the parts; if they are similar, we then compare the minimum X values, and finally, the minimum Y values if needed.  This "lexicographic" ordering is similar to the strategy used in autoregressive mesh generation models [*4, *5].
>
> > Canonical Positioning: Is the canonical position of each part defined based on the final assembled object, or can the object shape and part positions be learned independently?
>
> The canonical position and orientation of each part in our method are defined based on the final assembled object. Generating the object shape and parts independently would require a different design paradigm. While some scene generation works [*6, *7, *8] explore independent instance generation followed by placement, such approaches often suffer from inaccurate spatial relationships. In contrast, for part-level modeling, maintaining correct relative positions is crucial since parts must be composed into a coherent object. Defining part size, position, and orientation based on the assembled object allows us to preserve spatial consistency and eliminates the need for a separate optimization step during assembly.
>
> > Detail Level of Parts: How can we assess the level of detail in generated parts? For example, in the room scene (Fig. 1 middle), the “chair” part is generated as a whole, but generating a chair object as parts typically involves distinct components like handles and wheels.
>
> Thank you, this is an interesting question. In our current setup, the level of detail in part decomposition is implicitly determined by the granularity of artist-annotated parts in the training data. For example, if the full object is a chair, the decomposition typically includes parts like legs, cushion, and backrest. However, if the full asset contains multiple objects, such as chairs and tables, the decomposition often reflects object-level grouping. In practice, as in PartGen [5], our model can learn all such decompositions, if given in the training data, and can generate different ones simply by resampling. In addition, if the user wants a more fine-grained decomposition, the output part latent can be fed back into AutoPartGen as input (object-to-parts).
>
> In general, our model learns this granularity directly from the data distribution, and so a different granularity can be learned by tweaking the training data.
>
> In addition, we provide a masks-to-parts setting as described in the paper, where users can control the decomposition level using 2D masks. These masks can be manually annotated or generated using segmentation pipelines like SAM.
>
> > Necessity of Part-wise Generation: If the model is capable of generating full objects, what is the motivation for generating parts separately?
>
> Thank you. While both previous works and our method can generate a whole object, part-level generation is motivated by several practical and technical needs (L17-26):
> - **3D editing and customization**: In real-world applications, part-based generation enables intuitive editing. Users can modify or replace specific components (e.g., change the legs of a chair or the handle of a mug) without regenerating the entire object.
> - **Animation and articulation**: Many objects, especially those with moving parts (like doors, arms, or wheels), require separate motion for each part. Part-wise generation is essential for producing objects that can be easily rigged and animated.
> - **Robotic manipulation and spatial intelligence**: Robots often need to identify and interact with functional parts of objects, such as handles, lids, or levers. Explicit part representations make it easier for robots to understand object functionality and graspable regions. In addition, part-wise generation supports scalable data creation for robotics, as robots typically interact with objects at the part level during tasks like grasping, assembly, and tool use [*9, *10].
> - **Material assignment**: Different parts of an object often have different physical properties and materials (e.g., a chair with a metal frame and fabric seat). Part-wise generation makes it easier to assign materials in a semantically consistent way.
> - **More detailed generation**: Since each part is generated with different VecSet latents, the model can allocate capacity to capture fine-grained geometry per part. This leads to higher overall fidelity and expressiveness than generating a coarse full-object mesh in fixed-length VecSet.
> - **Fabrication, including 3D printing**: Many fabrication workflows, including 3D printing and physical assembly, require objects to be split into discrete parts. Part-wise generation aligns naturally with this process, enabling easier design, modification, and physical realization of objects.
>
>
> [*1] Chen, Yiwen, et al. "Ultra3D: Efficient and High-Fidelity 3D Generation with Part Attention." arXiv preprint arXiv:2507.17745 (2025).
>
> [*2] Bensadoun, Raphael, et al. "Meta 3d texturegen: Fast and consistent texture generation for 3d objects." arXiv preprint arXiv:2407.02430 (2024).
>
> [*3] Huang, Zehuan, et al. "Mv-adapter: Multi-view consistent image generation made easy." arXiv preprint arXiv:2412.03632 (2024).
>
> [*4] Lai, Zeqiang, et al. "Unleashing vecset diffusion model for fast shape generation." arXiv preprint arXiv:2503.16302 (2025).
>
> [*5] Siddiqui, Yawar, et al. "Meshgpt: Generating triangle meshes with decoder-only transformers." Proceedings of the IEEE/CVF conference on computer vision and pattern recognition. 2024.
>
> [*6] Wang, Kai, et al. "Planit: Planning and instantiating indoor scenes with relation graph and spatial prior networks." ACM Transactions on Graphics (TOG) 38.4 (2019): 1-15.
>
> [*7]  Nie, Yinyu, et al. "Total3dunderstanding: Joint layout, object pose and mesh reconstruction for indoor scenes from a single image." Proceedings of the IEEE/CVF Conference on Computer Vision and Pattern Recognition. 2020.
>
> [*8] Gao, Daoyi, et al. "Diffcad: Weakly-supervised probabilistic cad model retrieval and alignment from an rgb image." ACM Transactions on Graphics (TOG) 43.4 (2024): 1-15.
>
> [*9] Liu, Weiyu, et al. "Composable part-based manipulation." arXiv preprint arXiv:2405.05876 (2024).
>
> [*10] Huang, Haoxu, et al. "Copa: General robotic manipulation through spatial constraints of parts with foundation models." 2024 IEEE/RSJ International Conference on Intelligent Robots and Systems (IROS). IEEE, 2024.

---

### Official Review · Reviewer_bEVk · 2025-06-22

**Clarity:** 1
**Significance:** 3
**Originality:** 3
**Rating:** 3
**Confidence:** 4

**Summary:**

This work presents an auto-regressive method for 3D shape part generation. The approach dynamically decides which part to be generated and the number of generated parts. Results demonstrate it can achieve composable object generation without too much overlap.

**Questions:**

Basically, I encourage the authors to put more effort into improving the writing quality and paper organization. I did not think the paper is clear enough for readers to understand the technical details and reproduce the method.
The author mentioned the ambiguous problem for part generation. I wonder how the method addresses them.

**Ethical Concerns:**

["NO or VERY MINOR ethics concerns only"]

**Final Justification:**

Basically, I think it requires a careful revision and reconsideration about its core contributions. I do not object to its acceptance contingent on the following revisions.

1) A more focused contribution on diffusion-based AR models and their technical novelty.

2) A thorough revision for its technical method part to ease the understanding.

3) Evaluations on the role of diffusion parts and the advantage of compositional shape2vecset methods over other general ones. If there are other technical claims, a careful ablation over their alternatives should be evaluated.

**Limitations:**

It would be good to see discussions about failure cases.

**Paper Formatting Concerns:**

I did not see major formatting problems.

**Quality:**

2

**Strengths And Weaknesses:**

Strengths:
+ Generating part-level objects is essential for many downstream tasks.
+ The overall autoregressive generation is reasonable and novel.
+ The generated parts have less overlap and are of better quality.

Weakness:
- The paper and its motivation are not well organized. For example, I did not see why supporting different modal inputs is challenging in the design, but this is emphasized in the introduction. Moreover, I lost myself when going through technical details. For example, how to generate \hat{z}? If each step takes a point as input, this should be reflected in the pipeline, and there should be more descriptions about how the points are sampled. Line 162: Should there be an additional \hat{P} for the SDF function? It is unclear to me how the diffusion process is used. In Fig. 3, the symbol z^(1) is not defined, and there are inconsistent symbol usages for z in the method part. In line 181, is the equation correct? Is there any way to ensure the \hat{z} and the generated parts are consistent?
- The evaluation is not solid enough. If you have ground truth parts, why not evaluate the mIoU for all ablations and comparisons? For the present results, it would be hard for readers to judge the advantage of the method. Moreover, is there any specific reason why there is less overlap for the reported results?
- I did not see specific designs of AR models for part generations, and the shape generation is mainly based on Shape2VecSet.

---

> ### Author Rebuttal · Authors · 2025-07-29
>
> > I did not see why supporting different modal inputs is challenging in the design, but this is emphasized in the introduction.
>
> As in VecSet [59] and other prior works, each latent z is *generated* by a denoising diffusion model. The difference compared to prior work is how these generators are prompted (conditioned). In the original VecSet and other 3D generation works [18, 43, 59, 60, 61], the prompt is just an image. In our case, it can be an image, an image and a mask, and/or an encoded 3D shape, depending on which specific task one wants to solve. Hence, we design a unified model to handle different inputs in just one network. It is non-trivial that the same set of weights can generate parts across all these different scenarios and input modalities, as this requires training on diverse tasks using a unified training signal and supporting a general inference pipeline.
>
> > how to generate \hat{z}?
>
> To generate $\hat{z}$, users can either directly provide an unmasked image to our model or encode a 3D shape into $\hat{z}$. For example, one can use an existing 3D scan as shown in the paper (Section 4.2), or assets generated by other methods (L205-207).
>
> > If each step takes a point as input, this should be reflected in the pipeline, and there should be more descriptions about how the points are sampled. Line 162: Should there be an additional \hat{P} for the SDF function?
>
> We will clarify this process in the revision. As described in L151–154, the decoder is a function that takes a 3D point as input, along with the latent code (i.e., the encoded shape), and outputs the signed distance from the point to the shape surface. To extract the mesh, we evaluate this function over a grid of points and apply Marching Cubes to the resulting SDF values.
>
> > It is unclear to me how the diffusion process is used.
>
> The diffusion model is used to generate part latents, where each latent represents a distinct part of the object, conditioned on previous part latents and a choice of other modalities such as 3D shape, image, or masked image.
>
> > In Fig. 3, the symbol z^(1) is not defined, and there are inconsistent symbol usages for z in the method part.
>
> Thank you for the suggestion. We will revise the paper to clarify these notations. Briefly:
>
> - $z^{(i)}$: latent representation of the $i$-th part (L192).
> - $z$: latent representation of a part or the entire object, depending on context (L186–187).
> - $z^{(1,\dots,k-1)}$: latent representation of the combined mesh formed by parts 1 through $k-1$ (L203).
> - $\hat{z}$: latent representation of the whole object (L206).
> - $z_t$: noised version of $z$ used during the diffusion process (L179).
>
> > In line 181, is the equation correct?
>
> Thank you for spotting this! The equation should be $v(t, z_t,\epsilon) = (z_t - \sqrt{\alpha_t} \epsilon)/\sqrt{1 - \alpha_t}$. We have corrected this in the paper.
>
> > Is there any way to ensure the \hat{z} and the generated parts are consistent?
>
> We do this by explicitly conditioning the next part generation process on both the latent representation of previous generated parts $z^{(1,\dots,k-1)}$ and the whole object $\hat{z}$ (L205-211).
>
> > The evaluation is not solid enough. If you have ground truth parts, why not evaluate the mIoU for all ablations and comparisons? For the present results, it would be hard for readers to judge the advantage of the method.
>
> We have included quantitative results, including IoU, in our ablation study (see Tables 2 and 3 in the supplementary material). In addition, we have conducted additional experiments in response to other reviewers, which you can refer to.
>
> > Moreover, is there any specific reason why there is less overlap for the reported results?
>
> This is a result of our autoregressive design, as discussed in L295–298 and illustrated in Figure 8. Unlike parallel extraction, where parts are predicted independently, the autoregressive approach allows the model to capture dependencies between parts, helping to avoid undesirable overlaps. Theoretically, the autoregressive model learns the joint probability distribution over all parts, whereas the parallel approach models only the marginal distributions.
>
>
> > I did not see specific designs of AR models for part generations, and the shape generation is mainly based on Shape2VecSet.
>
> Thank you for the comment. While our shape representation builds on Shape2VecSet, we introduce three key architectural changes to enable autoregressive part generation:
> - **Autoregressive Diffusion Formulation:** We formulate the part generation process as an autoregressive diffusion model, which is novel in the context of 3D part generation. Unlike standard diffusion models that generate latent all at once, our approach generates each part sequentially, allowing the model to capture the dependencies between parts.
> - **Efficient Encoding of Previous Parts:** We explore how to best encode previously generated parts, as detailed in our additional ablation studies (see response to other reviewers). We propose a compact and efficient strategy to incorporate the information from earlier parts without increasing the token count or computational cost.
> - **Unified Formulation and Modeling:** We design a unified model and formulation that supports various input modalities, including 3D shapes, images, and masks, making the approach flexible and broadly applicable.

---

> ### Author Response · Authors · 2025-08-04
> **Looking forward to further discussions**
>
> Dear Reviewer bEVk,
>
> We would like to thank you for your constructive feedback.
>
> We hope our response has addressed your questions and concerns. Specifically, we have provided additional details about the method, included further comparisons between different conditioning strategies, and clarified both the motivation and the novelty of our work. We will further polish the writing and organization to improve clarity.
>
> As the author-reviewer discussion phase is ending soon, we would be happy to clarify any remaining points. If our response has sufficiently addressed your concerns, we would kindly appreciate it if you could consider updating your score.
>
> Thank you again for your time and thoughtful review.

---

> ### Author Response · Authors · 2025-08-05
> **Further response (Part 1)**
>
> Dear Reviewer bEVk,
>
> Thanks again for your constructive suggestion!
>
> > I reread the technical method and still found it to be kind of difficult to follow. I do not object to its acceptance contingent on a major revision of the paper, as well as the incorporation of important ablation studies in the main paper.
>
> We will revise the paper to improve clarity and presentation. We will also incorporate the ablation studies into the main paper, as suggested.
>
> > Based on the ablation, I think the major novelty lies in introducing the AR conditions, however, there is no specific design for the part generation task.
>
> As noted by the reviewer and detailed in both the paper and the rebuttal, we introduce a re-encoding strategy to make the autoregressive (AR) conditioning compact and efficient for part generation. **The re-encoding is a specific design** for the part generation task. During training, we define the position, scale, and orientation of each part relative to the overall object, rather than in a normalized space [*1, *2, *3]. This allows for straightforward spatial composition at the mesh level in re-encoding process. The re-encoding strategy allows our model to condition on a large number of previously generated parts while maintaining a nearly constant memory and computational footprint, which is crucial for scaling to complex objects or scenes. The experiments presented in the rebuttal (see responses to Reviewers V2su and kSwB) demonstrate both the effectiveness and efficiency of this approach. While we intentionally keep the generation process simple and general for scalability, it remains effective across various input settings and supports application like scene generation and city generation.
>
> **The overall generation pipeline is specifically designed for part generation** and employs a combination of auto-regressive modeling and diffusion. The auto-regressive generation process, along with our conditioning strategy, allows the model to flexibly handle varying numbers of parts, as different objects naturally contain different part counts. Our design also enables the generation of very long sequences, such as scene and city generation demonstrated in the paper. Diffusion is used to address the ambiguity inherent in the part generation task, and we provide a detailed discussion of this below.
>
> > Using an autoregressive diffusion model may be novel (I am not an expert on diffusion), but it requires the ablation of its role.
>
> Thank you for the comment. We have already provided an ablation of the autoregressive (AR) modeling, comparing it to non-AR variants where parts are generated in parallel/independently (i.e., diffusion models conditioned only on images, masks, and the overall shape, without conditioning on previously generated parts); see Lines 294–298, Figure 8 in the main text, and Table 2 in the supplementary materials. We therefore believe the reviewer is referring specifically to the role of the diffusion model within the AR generation process. In our framework, the diffusion model is used to handle both the ambiguity in part decomposition and the ambiguity arising from 2D-to-3D prediction. We agree that an explicit ablation of the diffusion component in this context would be helpful, and we will try to include it in the revision.
>
> [*1] Wang, Kai, et al. "Planit: Planning and instantiating indoor scenes with relation graph and spatial prior networks." TOG. 2019.
>
> [*2] Nie, Yinyu, et al. "Total3dunderstanding: Joint layout, object pose and mesh reconstruction for indoor scenes from a single image." CVPR. 2020.
>
> [*3] Gao, Daoyi, et al. "Diffcad: Weakly-supervised probabilistic cad model retrieval and alignment from an rgb image." TOG. 2024.

---

> > ### Comment · Reviewer_bEVk · 2025-08-07
> >
> > Thank you for your responses. I do not think it is a good practice to claim technical parts that are not the focus in the introduction and abstraction of the submitted version, e.g., the re-encoding strategy. Moreover, this re-encoding simply composes existing parts into a whole for encoding. I would say it's a design choice rather than a novel technical contribution. Maybe the diffusion designs for part generation are novel, but without the result support from the main paper, I was not convinced. Therefore, I think it is safe to say the main contribution is the introducing of autoregressive modelling, but there are no specific designs in the network structure and the way parts are generated.
> > I am not sure how we can enforce a proper major revision of the paper so that it will have clear technical claims, clear technical descriptions, and consistent experimental support.
> > Thank you.

---

> ### Author Response · Authors · 2025-08-05
> **Further response (Part 2)**
>
> > Though the authors claimed the unified modelling of different inputs, I did not see its difference from other generative models that take different inputs as conditions. Maybe there are some differences in the encoder network. The authors are encouraged to make the technical contribution of the paper clearer.
>
> Thank you for raising this point. We agree that unified conditional modeling is not the main technical contribution of our paper. However, we believe it is an important design choice that enhances the practicality and generality of our method.
>
> Unlike previous works that primarily focus on a single modality such as image or text for 3D generation [9, 28, 29, 45, 57, 59, 60, 61], and often require separate models for different input types [28, 59, 60], our framework adopts a unified architecture that flexibly supports multiple input modalities. Without any architectural changes or additional model instances, our model can seamlessly handle various conditioning settings, including masks-to-parts, image-to-parts, and object-to-parts, all using the same backbone.
>
> We believe this is particularly valuable for real-world deployment, as it eliminates the need to maintain multiple specialized models and promotes better generalization across tasks.
>
>
> > L139: I did not see why conditional probability can address the ambiguity.
>
> Conditional probability helps address two main sources of ambiguity in our problem:
>
> 1. **Ambiguity in Part Decomposition**: Part decomposition can vary from person to person. For example, one artist might split a chair into legs, seat, and backrest, while another might group the base and separate the cushion. There is no unique ground truth. Our model handles this by learning a conditional distribution over plausible decompositions. By sampling from this distribution, we can generate diverse and valid decomposition that reflect the inherent ambiguity.
>
> 2. **Ambiguity from Incomplete Conditions**: The input condition, such as a single 2D image (and masks), is often an incomplete representation of the full 3D object. For instance, the image provides no information about the object's back view or parts that are heavily occluded.  A deterministic model would have to guess a single, potentially inaccurate completion. In contrast, our probabilistic framework models a distribution over possible 3D shapes conditioned on the input, enabling it to generate multiple realistic completions and effectively handle uncertainty in the input.
>
> > Both AR models and diffusion models learn the joint distribution. "Theoretically, the autoregressive model learns the joint probability distribution over all parts, whereas the parallel approach models only the marginal distributions."
>
> To clarify, the “parallel approach” referenced here refer to prediciting the parts independently (in parallel) in the context of our rebuttal (*"Unlike parallel extraction, where parts are predicted independently, the autoregressive approach allows the model to capture dependencies between parts"*). The marginal distribution here means that the model conditions only on inputs such as 2D images, masks, or overall shapes, without explicitly modeling dependencies between parts.
>
> ---
> We hope our response has addressed your remaining concerns.
>
> If you have any further questions, please feel free to let us know. We would be happy to clarify any points during the discussion phase and will make every effort to respond promptly, given the limited time remaining.

---

> ### Author Response · Authors · 2025-08-07
> **Looking forward to your response**
>
> Dear Reviewer bEVk,
>
> We hope our latest response has addressed your questions and concerns during the discussion. As the author-reviewer discussion phase will end in two days, we would be happy to clarify any remaining points before then.
>
> If our response has sufficiently addressed your concerns, we would greatly appreciate it if you could consider updating your score.
>
> Thank you again for your time and thoughtful review.

---

> ### Author Response · Authors · 2025-08-08
>
> Thanks for your response.
>
> We would like to clarify that we did not present re-encoding as a core technical contribution. Rather, we present it as an efficient and effective task-specific design for part generation, supported by experiments (see response to Reviewer V2su, kSwB), in direct response to the reviewer’s request for specific designs tailored to this task.
>
> More broadly, we believe our paper offers meaningful contributions. That said, we agree with the reviewer’s suggestions and will revise the manuscript to make these contributions clearer by restructuring the introduction, method, and experiments. Specifically, we can highlight the following four contributions and map them clearly to empirical evidence:
>
> - Contribution 1: **Autoregressive 3D part generation.** We demonstrate the effectiveness of an autoregressive approach for 3D part generation. Unlike non-autoregressive methods that generate parts independently, our model accounts for dependencies between parts (see Lines 295–297, Figure 8, and Table 2 in the supplementary materials). This leads to reduced part overlap, automatic part separation, and improved generation quality. The re-encoding scheme was specifically designed for this setting, as noted by Reviewer kSwB, and its efficiency and effectiveness are demonstrated in the rebuttal (see response to Reviewer V2su, kSwB).
>
> - Contribution 2: **No reliance on external segmentation models.** Our model can identify and generate parts directly from input conditions (e.g., images or shapes) without relying on any external segmentation model (see Section 4.2, image-to-parts and object-to-parts tasks). This fully automatic capability is central to the design, and the name AutoPartGen reflects both the autoregressive and automatic nature of the approach. When given additional conditions such as masks, our method even outperforms approaches like HoloPart, which depend on ground-truth 3D segmentations (Section 4.3).
>
> - Contribution 3: **Unified model for diverse part generation and decomposition tasks.** We demonstrate that the same model (shared architecture and weights) can handle a variety of input conditions, including images, images with masks, and 3D shapes (section 4.2, 4.3, 4.5), without requiring architectural changes or multiple model instances. This unified design is neither obvious nor trivial, as prior methods often rely on separate models for different inputs [28, 59, 60]. We believe this unified design is an empirical contribution, demonstrating the practical value and generality of AutoPartGen across diverse input conditions.
>
> - Contribution 4: **Compositionality of VecSet representation:** As noted by Reviewer kSwB and di68, we are the first to demonstrate that the VecSet representation supports compositional generation (see Figure 3). This is not only helpful for part generation but also crucial for scaling to larger and more complex scenes as shown in the examples of scene and city generation.
>
> > Therefore, I think it is safe to say the main contribution is the introducing of autoregressive modelling, but there are no specific designs in the network structure and the way parts are generated.
>
> We respectfully disagree with the reviewer’s comment that our method has no specific designs in the network structure and the way parts are generated, for the following reasons:
>
> 1. In autoregressive modeling, the way information flows between steps is a key design. Our re-encoding strategy and the use of previously generated parts as conditioning inputs are deliberate and tailored for scalable 3D part generation.
> 2. As detailed in our previous responses, the entire generation pipeline is specifically designed for part generation, combining autoregressive modeling with diffusion. This setup enables the model to flexibly handle varying numbers of parts. It also supports the generation of very long sequences, as demonstrated in our scene and city generation examples. Diffusion here plays a key role in addressing the ambiguity inherent in part generation, as discussed earlier.
>
> > Maybe the diffusion designs for part generation are novel, but without the result support from the main paper, I was not convinced.
>
> We briefly experimented with autoregressive modeling **without** diffusion, but found it very difficult to train; it showed no signs of convergence even after 100 epochs on 128 GPUs.

---

> > ### Comment · Reviewer_bEVk · 2025-08-09
> >
> > Thank you for the explanations. Maybe the authors should focus on the technical contribution of diffusion-based AR models for part generation. The second and third contributions may be the capability of the method rather than technical contributions. Claiming the fourth contribution requires more analysis. For example, if there are any quantitative measures to convince readers of what scales and complexity the decomposition applies to, what is the impact of taking some other autoencoder methods that do not have decomposition capability? In summary, the AR-based generation looks interesting and deserves more exploration, and the paper needs more careful revision to isolate the core contribution.

---

> > > ### Author Response · Authors · 2025-08-09
> > >
> > > Thank you for your follow-up comments. We agree that the core technical contribution is the integration of diffusion within an autoregressive framework for 3D part generation, and we will revise the paper to make this the central focus.
> > >
> > > > what is the impact of taking some other autoencoder methods that do not have decomposition capability?
> > > we agree this is worth investigating and plan to explore alternatives such as the autoencoder used in TRELLIS.
> > >
> > > Finally, we appreciate that you recognize the novelty and interest of our AR-based generation framework.

---

### Official Review · Reviewer_di68 · 2025-06-30

**Clarity:** 3
**Significance:** 3
**Originality:** 3
**Rating:** 5
**Confidence:** 4

**Summary:**

The method is about generating 3D objects/scenes composed of many parts. The training of the method contains two stages: 1) part autoencoder; 2) conditioned diffusion model. When doing inference, an object is generated part-by-part autoregressively. Extensive experiments are done to prove the effectiveness of the method, including object- and scene-level generation.

**Questions:**

I still do not understand how the parts are ordered. Usually, we need an order for the elements used in autoregressive models, like the natural language order or rasterization order for images. However, from the context, I do not get what the order is here.

The inference includes a (autoregressive) set of diffusion generation processes. I would expect it would be very slow. It would be better if there were a time analysis of the inference cost.

**Ethical Concerns:**

["NO or VERY MINOR ethics concerns only"]

**Final Justification:**

Thanks for the response. I recommend acceptance. The authors delivered some interesting results.

**Limitations:**

The method takes really lots of GPUs (128 for the autoencoder and 256 for the diffusion models) to train. This could be a major limitation for most universities (which do not have access to large computing clusters). However, this is not a real drawback. On the contrary, it shows the scaling ability.

**Paper Formatting Concerns:**

The references in the supplemental are not aligned with the main pdf.

**Quality:**

3

**Strengths And Weaknesses:**

The writing is clear. The core intuition is well-described in the introduction. It is based on a simple observation of the combinatorial property of 3DShape2VecSet (see below). The outer level of the generation is done in an autoregressive way, i.e., part-by-part. The inner level is done using diffusion models in the latent space of parts.

The property described in Fig 3 and Sec 3.1 is a very interesting observation. This not only helps the part-aware generation as shown in this paper, but also might be helpful in scaling 3DShape2VecSet to a larger latent space while preserving high-quality details. Actually, I am quite surprised about the property. It would be better if there were some math analysis about the property.

The main experiments are about object-level generation, including different conditions (images and objects). The experiment setup makes sense and the results are strong. There are also applications on scene-level generation.

---

> ### Author Rebuttal · Authors · 2025-07-29
>
> > The property described in Fig 3 and Sec 3.1 is a very interesting observation. This not only helps the part-aware generation as shown in this paper, but also might be helpful in scaling 3DShape2VecSet to a larger latent space while preserving high-quality details. Actually, I am quite surprised about the property. It would be better if there were some math analysis about the property.
>
> Thank you, we also find this property very interesting. As to why the compositionality works, this appears to be an emerging property of VecSet itself as we do not do special training to achieve it (which, we think, makes it even more interesting). The likely reason is that the transformer is trained to operate on sets -- for example, it is permutation equivariant -- and that the elements in these sets are actually "semantically" close to 3D points because they start as such in the encoder. So, just like the union of sets of 3D points is well defined, the union of these sets of code vectors is also meaningful to the model. Note that this is non-trivial, as the concatenation of the latent codes must then be decoded in a new, overall consistent SDF function for the full object.
>
> A recent work [*1] also has a related observation, as it found that the spatial queries of the decoder and shape latents show strong locality. Specifically, they find that nearby queries have high attention to a similar small group of latent tokens, which are then further used to accelerate the decoding process.
>
> > I still do not understand how the parts are ordered. Usually, we need an order for the elements used in autoregressive models, like the natural language order or rasterization order for images. However, from the context, I do not get what the order is here.
>
> Thank you for your helpful suggestion. In the masks-to-parts scenario, the order of the input masks determines the order in which parts are generated. In the automatic setting, we use a predefined order during training, which the model also follows during inference. Specifically, the training assets are defined in a canonical space, and parts are sorted based on their bounding boxes in the following order: bottom to top (Z), left to right (X), and front to back (Y), following Blender’s ZXY axis convention. We first compare the minimum Z values of the parts; if they are similar, we then compare the minimum X values, and finally, the minimum Y values if needed. This "*lexicographic*" ordering is similar to the strategy used in autoregressive mesh generation models [*2, *3].
>
>
> > The inference includes a (autoregressive) set of diffusion generation processes. I would expect it would be very slow. It would be better if there were a time analysis of the inference cost.
>
> The inference time of AutoPartGen primarily depends on the number of parts being generated, increasing approximately linearly since the time cost remains constant per part. We report a detailed time analysis in the table below. Not that each part generation involves several sequential steps: (1) encoding the input conditions (e.g., images and previously generated shape); (2) running latent diffusion; (3) decoding the latent into an SDF grid; and (4) extracting the mesh using Marching Cubes.
>
> | Encode Conditions | Diffusion  | Part Decoding |  Marching Cubes  |
> |-------|-------|-------|-------|
> |0.13s| 12.2s| 36.3s| 2.1s|
>
> As shown in the table, the current bottleneck of the pipeline lies in the diffusion process and decoding. To accelerate inference, we can adopt FlashVDM [*1], which reduces the decoding runtime from 36.3s to <10s. Additionally, efficiency can be further improved by employing a distillation strategy in the diffusion process, as demonstrated in [*1].
>
> > The method takes really lots of GPUs (128 for the autoencoder and 256 for the diffusion models) to train. This could be a major limitation for most universities (which do not have access to large computing clusters). However, this is not a real drawback. On the contrary, it shows the scaling ability.
>
> We agree that our training requires substantial resources (128 GPUs for the autoencoder and 256 for the diffusion model), which may not be accessible to all institutions. However, our goal is to demonstrate the scaling potential of the method, especially for highly detailed 3D generation and scene generation. Smaller-scale versions can be trained with fewer resources by reducing model and batch sizes, though this typically comes at the cost of reduced performance. We briefly experimented with training on 64 GPUs and found that the model converges more slowly, with a ~0.03 IoU drop compared to the model trained with the same number of iterations on 256 GPUs.
>
> [*1] Lai, Zeqiang, et al. "Unleashing vecset diffusion model for fast shape generation." arXiv preprint arXiv:2503.16302 (2025).
>
> [*2] Siddiqui, Yawar, et al. "Meshgpt: Generating triangle meshes with decoder-only transformers." Proceedings of the IEEE/CVF conference on computer vision and pattern recognition. 2024.
>
> [*3] Chen, Yiwen, et al. "Meshanything: Artist-created mesh generation with autoregressive transformers." arXiv preprint arXiv:2406.10163 (2024).

---

### Official Review · Reviewer_kSwB · 2025-07-03

**Clarity:** 3
**Significance:** 3
**Originality:** 3
**Rating:** 4
**Confidence:** 5

**Summary:**

This paper proposes a new part-based 3D generator, named AutoPartGen. AutoPartGen leverages the locality property of the vecset-based 3D VAE latent space and trains a flow model to autoregressively generate part latents. The authors conduct several experiments to validate the proposed method.

**Questions:**

See the weaknesses, especially the part-order ambiguity.

**Ethical Concerns:**

["NO or VERY MINOR ethics concerns only"]

**Final Justification:**

Most of my concerns are addressed by the rebuttal. Therefore, I tend to maintain my score.

**Limitations:**

Yes

**Quality:**

3

**Strengths And Weaknesses:**

Strengths:

1. The observation that locality, or at least compositionality, is well supported by the vecset representation is insightful. Interestingly, the authors demonstrate that directly concatenating two different latents can be decoded together.

2. The proposed autoregressive generation process is reasonable. Additionally, AutoPartGen re-encodes all previously generated meshes to obtain more compact conditional latents, enhancing efficiency.

3. The presentation is clear and easy to follow.

Weaknesses:

1. AutoPartGen suffers from part order ambiguity. The authors do not clarify how the model defines the autoregressive order. From my perspective, the conditionally masked image $J$ could provide some information on which part to generate. However, the masked image may contain limited information when the part size is small.

2. More ablation experiments are expected, particularly regarding the re-encoding strategy to inject previous information.

3. It is unclear how AutoPartGen generates texture, as the vecset VAE does not encode appearance information.

---

> ### Author Rebuttal · Authors · 2025-07-28
>
> > AutoPartGen suffers from part order ambiguity. The authors do not clarify how the model defines the autoregressive order. From my perspective, the conditionally masked image $J$ could provide some information on which part to generate. However, the masked image may contain limited information when the part size is small.
>
> Thank you for your helpful suggestion. In the masks-to-parts setting, the order of the input masks determines the order in which parts are generated. As described in the paper, we provide both the unmasked and masked images to help the model learn the spatial relationship between the parts and the whole object in 3D space. In our experiments, we found that the model can accurately infer very small parts (less than 0.1% of the object’s volume) with very small masks. In addition, the users could interactively adjust the image guidance scale to control the effect of masks and images (L213-217).
>
> In the automatic setting, we use a predefined order during training, which the model also follows during inference. Specifically, the training assets are defined in a canonical space, and parts are sorted based on their bounding boxes in the following order: bottom to top (Z), left to right (X), and front to back (Y), following Blender’s ZXY axis convention. We first compare the minimum Z values of the parts; if they are similar, we then compare the minimum X values, and finally, the minimum Y values if needed. This "*lexicographic*" ordering is similar to the strategy used in autoregressive mesh generation models [*1, *2].
>
> We have revised the paper accordingly to include more details about the part order.
>
> > More ablation experiments are expected, particularly regarding the re-encoding strategy to inject previous information.
>
> Thanks! We have conducted additional experiments to further explore this direction. Specifically, we compare the re-encoding strategy with two alternatives: simply concatenating all part tokens, and using a latent fuser module based on a 6-layer Perceiver-style network that compresses the concatenated VecSet into a fixed-length latent. The performance comparison between these three approaches is shown in the table below. All models are trained for 150 epochs using 512 tokens per part under the same training setup. As shown in the table, all three methods achieve similar results, with re-encoding performing slightly better than the other two methods.
>
>
> | Method | IoU ↑ | F-Score ↑ | Chamfer Distance ↓|
> |-|-|-|-|
> | Re-encode |  0.627 | 0.815 | 0.055 |
> | Concat | 0.611 | 0.804 | 0.059 |
> | Latent Fuser |  0.608 | 0.802 | 0.061 |
>
> We also provide GPU memory and runtime analysis for the different approaches, as shown below. Decoding time is not included, since all methods use the same mesh decoding process for each part. The experiments are conducted on a single NVIDIA A100 GPU (80GB) using PyTorch 2.7.1 with F.scaled_dot_product_attention.
>
> | #Part | Method  | Inference GPU Memory (GB) | Training GPU Memory (GB) | Inference Speed (sec/step) | Training Speed (sec/step) |
> |-------|----------------|---------------------------|---------------------------|-----------------------------|----------------------------|
> | 5     | Re-encode      | 17.9                      | 58.6                      | 0.616                       | 1.101                      |
> |       | Concat         | 30.3                      | 58.8                      | 0.737                       | 1.378                      |
> |       | Latent Fuser   | 30.3                      | 58.6                      | 0.622                     | 1.181                      |
> | 10    | Re-encode      | 17.9                      | 58.6                      | 0.626                       | 1.101                      |
> |       | Concat         | 33.3                      | 63.6                      | 0.956                       | 1.845                      |
> |       | Latent Fuser   | 33.3                      | 58.6                      | 0.644                       | 1.303                      |
> | 30    | Re-encode      | 17.9                      | 58.6                      | 0.626                       | 1.101                      |
> |       | Concat         | 45.4                      | OOM                       | 1.826                       | OOM                        |
> |       | Latent Fuser   | 45.3                      | 58.6                      | 0.736                       | 1.799                      |
>
> From the table above, we observe that concatenating all latents (Concat) significantly increases memory usage and slows down training, even when using the latest efficient PyTorch implementation. This is mainly because the number of tokens grows rapidly with the number of parts. To address this, we adopt the re-encoding strategy (Re-encode), where both runtime and memory usage remain stable even with a large number of parts, since the mesh joining step is negligible in time and GPU memory. Regardless of how many parts are involved, they are re-encoded into a fixed-length VecSet.
>
> > It is unclear how AutoPartGen generates texture, as the vecset VAE does not encode appearance information.
>
> Similar to VecSet [59] and other related works [28, 29, 55, 60, 61, *1, *2], our model focuses solely on generating 3D geometry without textures. The primary focus of this paper is on modeling shape, and the VecSet representation does not include appearance information. However, VecSet is typically used in conjunction with texture generation models such as TextureGen [*3] or MV-Adapter [*4], as demonstrated in [28, 29, 60], to generate appearance after geometry is established. We plan to further explore this direction in future work, aiming to jointly model both geometry and appearance.
>
> [*1] Siddiqui, Yawar, et al. "Meshgpt: Generating triangle meshes with decoder-only transformers." Proceedings of the IEEE/CVF conference on computer vision and pattern recognition. 2024.
>
> [*2] Chen, Yiwen, et al. "Meshanything: Artist-created mesh generation with autoregressive transformers." arXiv preprint arXiv:2406.10163 (2024).
>
> [*3] Bensadoun, Raphael, et al. "Meta 3D TextureGen: Fast and consistent texture generation for 3d objects." arXiv preprint arXiv:2407.02430 (2024).
>
> [*4] Huang, Zehuan, et al. "MV-Adapter: Multi-view consistent image generation made easy." arXiv preprint arXiv:2412.03632 (2024).

---

> > ### Comment · Reviewer_kSwB · 2025-08-02
> >
> > Thanks for the response. All of my concerns are addressed. I tend to keep my score.

---

> > > ### Author Response · Authors · 2025-08-04
> > >
> > > Thank you again for your constructive feedback. We’re glad that our response has addressed your concerns.

---

### Official Review · Reviewer_V2su · 2025-07-03

**Clarity:** 1
**Significance:** 2
**Originality:** 3
**Rating:** 4
**Confidence:** 5

**Summary:**

This paper introduces an autoregressive model for generating compositional 3D objects part by part. To achieve this, the authors first parameterize 3D objects into a latent space using the existing 3DShape2VecSet representation. They then train a latent diffusion model to progressively predict the latent code of each 3D part, conditioned on input evidence and the latents of previously generated parts. While the experimental results demonstrate plausible generation outcomes, the proposed method does not present significant technical innovation. Additionally, the overall writing quality of the paper could be improved.

**Questions:**

Here are my specific questions and suggestions:

1. In the Introduction, it would be helpful to provide a visual example to illustrate the statement that "the concatenation of two codes can be decoded into the union of the corresponding surfaces." Moreover, although the importance of concatenation is emphasized, the actual implementation does not use concatenation but instead re-encodes the known parts. What is the purpose of discussing concatenation in the Introduction if it is not used in practice?

2. The paper claims that the method can automatically generate 3D objects part by part, but does not describe a termination strategy. How does the model determine when the scene has been fully generated?

3. In Sec 3.1, the function ‘sample_N x’ would be clearer if written as ‘sample(N, x)’ to indicate its inputs. Additionally, the point set P is sampled from object x, and then ~P is sampled from P. Why not sample M points directly from x? What is the rationale behind this two-step sampling process?

4. What is the architecture of the decoder D? How is it trained? What is the underlying principle that enables it to achieve the results shown in Fig. 3? Please provide a detailed discussion and explanation.

5. In Sec 3.3, the process involves decoding, sampling, merging, and re-encoding latent codes. How does this approach differ from simply concatenating the latents? It seems that encoding a small 3D part and encoding a complex scene with the same encoder may yield different results. Please provide comparative experiments to demonstrate the differences between these two strategies.

6. How are different parts defined in the training data? How do you ensure that each part has a consistent semantic meaning?

7. The writing, especially in the Methods section, is somewhat verbose and difficult to follow. I recommend revising and refining this section to more clearly define key concepts and articulate the core methodology.

8. Missing citations or comparisons:
- MIDI: Multi-Instance Diffusion for Single Image to 3D Scene Generation
- Zero-Shot Scene Reconstruction from Single Images with Deep Prior Assembly
- CAST: Component-Aligned 3D Scene Reconstruction from an RGB Image
- Part123: Part-aware 3D Reconstruction from a Single-view Image

9. Minor Issues and Typos:
- The function ‘sample_N’ is defined multiple times in Sec 3.3.
- The pipeline shown in Fig 2 is not referenced or explained in the main text.

**Ethical Concerns:**

["NO or VERY MINOR ethics concerns only"]

**Final Justification:**

The authors' response has largely addressed my concerns. I agree to accept the paper contingent upon the promised content being incorporated into the revised camera-ready version.

**Limitations:**

No, in the conclusion, the authors need to provide a necessary discussion of the limitations of the method.

**Quality:**

2

**Strengths And Weaknesses:**

Strengths:
The method achieves impressive results in generating compositional 3D objects.

Weaknesses:
The technical novelty of the paper is limited, as the approach essentially fine-tunes a latent diffusion model. Besides, many technical details are unclear, and the paper lacks necessary ablation studies and relevant citations.

---

> ### Author Rebuttal · Authors · 2025-07-28
>
> > The technical novelty of the paper is limited, as the approach essentially fine-tunes a latent diffusion model. Besides, many technical details are unclear, and the paper lacks necessary ablation studies and relevant citations.
>
> We respectfully disagree with the point and would like to summarize our contributions as follows:
>
> - **First to explore auto-regressive 3D part generation**, demonstrating the potential of auto-regressive generation for scalability and broader applications such as scene and city generation. Other works on part generation are *not* autoregressive. We specifically designed the re-encoding schema for efficiency as noted by Reviewer kSwB.
>
> - **No reliance on external 2D segmentation models**: Unlike most prior works, our method enables the model to learn 3D part decomposition directly, without additional 2D masks.
>
> - **Unified model for diverse part generation and decomposition tasks**: We show that a single model can generalize across a wide range of decomposition tasks, taking as input different modalities (3D models, images, and masks). No other part generator is this flexible.
>
> - **Compositionality of VecSet representation**: As noted by Reviewer kSwB and di68, we are the first to demonstrate that the VecSet representation supports compositional generation. This is not only helpful for part generation but also crucial for scaling to larger and more complex scenes, as shown in the examples of scene and city generation.
>
> > In the Introduction, it would be helpful to provide a visual example to illustrate the statement that "the concatenation of two codes can be decoded into the union of the corresponding surfaces." Moreover, although the importance of concatenation is emphasized, the actual implementation does not use concatenation but instead re-encodes the known parts. What is the purpose of discussing concatenation in the Introduction if it is not used in practice?
>
> Thank you for your suggestion! We have provided a visual example in Figure 3 and we plan to include more examples and discussion in the revision. The point is to show that the VecSet representation has good local properties which are suitable for part generation as each part is represented by an independent VecSet. However, instead of naively concatenating part codes, we opted for re-encoding the previously generated parts into a fixed-length VecSet, which allows us to bound the number of tokens entering cross-attention (lines 201–203) and results in improved efficiency. In particular, when the number of parts is large (the city generation example), simple concatenation would lead to a huge computation cost and would easily go out of memory.
>
> To further explore this idea, we also experimented with a latent fuser module (a 6-layer Perceiver-style network) that explicitly fuses the *concatenated* VecSet tokens into a fixed-length latent. We provide the performance comparison in the following table. All models are trained for 150 epochs with 512 tokens per part and the same training setup. As shown in the table, all three methods achieve similar results, with re-encoding performing slightly better than the other two methods.
>
> | Method | IoU ↑ | F-Score ↑ | Chamfer Distance ↓|
> |-|-|-|-|
> | Re-encode |   0.627 | 0.815 | 0.055 |
> | Concat | 0.611 | 0.804 | 0.059 |
> | Latent Fuser |  0.608 | 0.802 | 0.061 |
>
> We also provide GPU memory and runtime analysis for the different approaches, as shown below. Decoding time is not included, since all methods use the same mesh decoding process for each part. The experiments are conducted on a single NVIDIA A100 GPU (80GB) using PyTorch 2.7.1 with F.scaled_dot_product_attention.
>
> | #Part | Method  | Inference GPU Memory (GB) | Training GPU Memory (GB) | Inference Speed (sec/step) | Training Speed (sec/step) |
> |-------|----------------|---------------------------|---------------------------|-----------------------------|----------------------------|
> | 5     | Re-encode      | 17.9                      | 58.6                      | 0.616                       | 1.101                      |
> |       | Concat         | 30.3                      | 58.8                      | 0.737                       | 1.378                      |
> |       | Latent Fuser   | 30.3                      | 58.6                      | 0.622                     | 1.181                      |
> | 10    | Re-encode      | 17.9                      | 58.6                      | 0.626                       | 1.101                      |
> |       | Concat         | 33.3                      | 63.6                      | 0.956                       | 1.845                      |
> |       | Latent Fuser   | 33.3                      | 58.6                      | 0.644                       | 1.303                      |
> | 30    | Re-encode      | 17.9                      | 58.6                      | 0.626                       | 1.101                      |
> |       | Concat         | 45.4                      | OOM                       | 1.826                       | OOM                        |
> |       | Latent Fuser   | 45.3                      | 58.6                      | 0.736                       | 1.799                      |
>
> From the table above, we observe that concatenating all latents (Concat) significantly increases memory usage and slows down training, even when using the latest efficient PyTorch implementation. This is mainly because the number of tokens grows rapidly with the number of parts. To address this, we adopt the re-encoding strategy (Re-encode), where both runtime and memory usage remain stable even with a large number of parts, since the mesh joining step is negligible in time and GPU memory. All parts are re-encoded into a fixed-length VecSet regardless of their number.
>
> > The paper claims that the method can automatically generate 3D objects part by part, but does not describe a termination strategy. How does the model determine when the scene has been fully generated?
>
> This is described in L210-211: the model terminates by generating the [EoT] token, which we simply define to be a zeroed-out shape vector $z$.
>
> >  In Sec 3.1, the function ‘sample_N x’ would be clearer if written as ‘sample(N, x)’ to indicate its inputs. Additionally, the point set P is sampled from object x, and then ~P is sampled from P. Why not sample M points directly from x? What is the rationale behind this two-step sampling process?
>
> As described in L158-159, the VecSet representation encodes all the $N$ points -- it randomly selects a smaller subset $M \subset N$ of these $N$ points as a "support set" ("query") as the input of encoder, but then encodes *all* the $N$ points via cross-attention (a component in the encoder function). This is just how the standard VecSet encoder is defined [29,55,59,60].
>
> > What is the architecture of the decoder D? How is it trained? What is the underlying principle that enables it to achieve the results shown in Fig. 3? Please provide a detailed discussion and explanation.
>
> As detailed in L846-858 in the supplementary material, the decoder architecture is similar to the 3DShape2VecSet [59], with the major difference of a larger model, training loss, data augmentation and the use of an SDF decoder instead of an opacity decoder. As to why the compositionality works, this appears to be an emerging property of VecSet itself, as we do not do special training to achieve it (which, we think, makes it an interesting observation). The likely reason is that the transformer is trained to operate on sets -- for example, it is permutation equivariant -- and that the elements in these sets are actually "semantically" close to 3D points because they start as such in the encoder. So, just like the union of sets of 3D points is well defined, the union of these sets of code vectors is also meaningful to the model. Note that this is non-trivial, as the concatenation of the latent codes must then be decoded in a new, overall consistent SDF function for the full object.
>
> > How are different parts defined in the training data? How do you ensure that each part has a consistent semantic meaning?
>
> We define parts in the same way as PartGen [5]: these are artist-like parts, where the semantics are implicitly derived from how 3D artists naturally decompose objects. As described in Lines 261–268, each artist-created asset is stored in glTF/GLB format, which contains multiple meshes (i.e., parts) along with the hierarchical structure as defined by the artist. In practice, this setup captures a diverse range of plausible decompositions, which our model can sample from due to its probabilistic nature.
>
> > The writing, especially in the Methods section, is somewhat verbose and difficult to follow. I recommend revising and refining this section to more clearly define key concepts and articulate the core methodology.
>
> Thank you, we will revise the method section to include more details.
>
> > Missing citations or comparisons.
>
> Thank you for the constructive suggestions. We will cite all the recommended papers accordingly. We will add a detailed discussion section with related works. In fact, Part123 is already cited as [31]. The scope of MIDI, Deep Prior Assembly, and CAST differs from ours, as these works primarily focus on the instance level rather than the part level, making direct comparisons difficult. Furthermore, CAST was released on arXiv in February and is not open-sourced yet. As for Part123, the method only infers surface decompositions without producing complete parts, and it relies on an external 2D segmentation model such as SAM. In addition, MIDI is limited to generating up to five instances due to its reliance on costly attention between latent codes, whereas our method can efficiently generate tens of part tokens.
>
> > Minor Issues and Typos.
>
> Thanks! We will correct them in the revision.

---

> > ### Comment · Reviewer_V2su · 2025-08-04
> >
> > Thank you to the authors for the response; it has addressed most of my concerns regarding the paper. However, I still have some doubts about the necessity of the two-step sampling process mentioned in Q3. Additionally, I am curious about how, after re-encoding, the features of the generated parts—which undergo further compression—can still reliably represent the combined object. I suggest the authors provide visual results demonstrating the effect of the ‘re-encoding’ process, similar to those shown in Fig. 3. In light of your reply, I am willing to revise my rating to Borderline Reject. If the authors could provide a clear and reasonable explanation on these points, I would be happy to reconsider and potentially update my rating further.

---

> > > ### Author Response · Authors · 2025-08-05
> > > **Response to further questions (Part 1)**
> > >
> > > We sincerely appreciate your timely response. Please find our answers to your follow-up questions below.
> > >
> > > > I still have some doubts about the necessity of the two-step sampling process mentioned in Q3.
> > >
> > > Thank you for raising this point. To further clarify, as described in both the paper and the rebuttal, all $M$ points are used in the encoding process via cross-attention. We therefore believe the reviewer is referring to whether the VAE’s reconstruction ability is affected by sampling $N$ points directly from the shape $x$ (“Direct Sampling”) versus sampling from the pre-sampled $M$ points (“Two-step Sampling” or “Sampling from $M$”).
> > >
> > > We provide an additional comparison between these two strategies in the following table. To compute the metrics, we use 512 meshes in the test set, and IoU is computed using a grid size of 256.
> > >
> > > | Strategy | IoU ↑ | F-Score ↑ | Chamfer Distance ↓|
> > > |-|-|-|-|
> > > | Direct sampling |  0.836 | 0.975 | 0.0188 |
> > > | Sampling from $M$ | 0.839 | 0.978 | 0.0185 |
> > >
> > > According to the table, we observe that sampling points from the pre-sampled $M$ points and directly sampling from the original shape $x$ result in very similar performance, with sampling from $M$ points giving slightly better results. This is somewhat expected, as the $N$ sampled points only serve as spatial queries. Since we apply farthest point sampling (FPS) in both strategies following [59], the selected points should cover the overall shape well, regardless of whether they are sampled from $x$ or $M$. Additionally, the $M$ points provide richer geometric details through cross-attention during the encoding process.
> > >
> > > In our implementation, we choose to sample from $M$ points to align with common practice in related work [29, 55, 59, 60]. The time difference between the two strategies is negligible, with both taking less than 0.05 seconds.
> > >
> > > In the original VecSet paper [59], the authors also experimented with learned queries, but found that subsampled queries from $M$ points achieved slightly better performance.

---

> > > > ### Author Response · Authors · 2025-08-05
> > > > **Response to further questions (Part 2)**
> > > >
> > > > > Additionally, I am curious about how, after re-encoding, the features of the generated parts—which undergo further compression—can still reliably represent the combined object. I suggest the authors provide visual results demonstrating the effect of the ‘re-encoding’ process, similar to those shown in Fig. 3.
> > > >
> > > > Thank you for the suggestion. We agree that visual illustrations of the re-encoding process would help clarify its effect. However, the NeurIPS policy this year explicitly prohibits the inclusion of external links or embedded images (e.g., base64 encoding) in the rebuttal, due to concerns about identity leakage and policy violations. As a result, we are unfortunately unable to include additional visualizations at this stage. Nevertheless, we agree that such examples would enhance clarity, and we will include them in the revised version of the paper to better illustrate the effectiveness of the re-encoding process.
> > > >
> > > > To further clarify, we provide a more detailed description of the re-encoding process below:
> > > > 1. Decode the current part latent into a grid of SDF values, then apply Marching Cubes to extract the corresponding mesh.
> > > > 2. Merge this part mesh with the previously generated part meshes to form a combined mesh.
> > > > 3. Encode the combined mesh back into a compact latent representation (2048 tokens in our case) using our VAE encoder.
> > > >
> > > > The combination happens at the mesh level and introduces negligible loss in fidelity or time cost. The fixed-length token representation effectively preserves most of the information from all previously generated parts.
> > > >
> > > > In the rebuttal, we have shown that re-encoding is an effective and efficient strategy to provide the model with information about previously generated parts, through experiments comparing different conditioning strategies. To further demonstrate the reliability of the re-encoding process, we compare the following three setups:
> > > > 1. Re-encode Combined Decoded Parts (Re-encode): Decode part and merge it with previously decoded parts into a combined mesh, then re-encode and decode this combined mesh. (This corresponds to our re-encoding strategy.)
> > > > 2. Encode Ground Truth Combined Mesh (Encode GT Combine): Directly encode the ground truth combined mesh and decode the latent to obtain the reconstructed mesh. (This serves as an upper bound, as it uses the ground truth mesh as input and primarily reflects the capacity of the VAE.)
> > > > 3. Separate Encode-Decode (Encode-Decode Separate Parts): Encode each part individually, decode them separately, and then merge the resulting meshes. (This corresponds to using a much larger latent representation, as the total number of tokens scales with the number of parts.)
> > > >
> > > > We evaluate the reconstructed combined meshes against the ground truth combined mesh using 512 meshes from the test set. For each mesh, we randomly select a certain number of parts while ensuring that all three setups share the same sampling configuration. Results are summarized below:
> > > >
> > > > |  | IoU ↑ | F-Score ↑ | Chamfer Distance ↓|
> > > > |-|-|-|-|
> > > > | Re-encode |  0.824 | 0.968 | 0.0151 |
> > > > | Encode GT Combined | 0.827 | 0.971 | 0.0148 |
> > > > | Encode-Decode Separate Parts | 0.853 | 0.976 | 0.0142 |
> > > >
> > > > As shown in the table, Re-encode is a reliable strategy for representing all previously generated parts. It achieves performance comparable to encoding the ground truth combined mesh, while separately encoding and decoding each part achieves slightly better results but requires a significantly larger number of tokens.
> > > >
> > > > We hope our response has addressed your remaining concerns.
> > > >
> > > > If you have any further questions, please feel free to let us know. We would be happy to clarify any points during the discussion phase and will make every effort to respond promptly, given the limited time remaining.

---

> > > > > ### Comment · Reviewer_V2su · 2025-08-05
> > > > >
> > > > > Thanks for the authors' response. I will change my rating to Borderline Accept.

---

> ### Author Response · Authors · 2025-08-04
> **Looking forward to further discussions**
>
> Dear Reviewer V2su,
>
> We would like to thank you for your constructive feedback.
>
> We hope our response has addressed your questions and concerns. Specifically, we have provided additional details about the method, included further comparisons between different conditioning strategies, and clarified both the motivation and the novelty of our work.
>
> As the author-reviewer discussion phase is ending soon, we would be happy to clarify any remaining points. If our response has sufficiently addressed your concerns, we would kindly appreciate it if you could consider updating your score.
>
> Thank you again for your time and thoughtful review.

---

### Note · Authors · 2025-08-12

We thank the ACs and reviewers for the thoughtful discussion.

To clarify, our **core contribution** is formulating 3D part generation as an autoregressive diffusion process, supported by a unified model that accepts diverse input modalities, including images, masks, and shapes, along with a re-encoding design for compact and efficient part generation. Our proposed **AutoPartGen** demonstrates **superiority** (outperforming state-of-the-art methods even with weaker inputs), **scalability** (ranging from object to scene and city generation), **generalization ability** (handling diverse modality inputs within a single model), and **flexibility** (supporting varying numbers of parts as well as different decompositions of the same asset).


We appreciate the reviewers’ recognition of our work’s strengths:
1. Strong results and high quality (V2su, bEVk)
2. Clear presentation and writing (kSwB, di68)
3. Interesting compositionality of the VecSet representation (kSwB, di68)
4. Reasonable and novel autoregressive diffusion formulation (kSwB, bEVk, di68, Hib7)
5. Flexibility across tasks and inputs (Hib7)

In the rebuttal and discussion phase, we addressed the reviewers’ concerns point by point:
1. Clarification of method details: Part ordering (canonical Z–X–Y) and termination ([EoT]); definition and usage of $\hat{z}$; part definition; role of diffusion; decoder design; and annotations.
2. Experiments and evidence: Additional conditioning ablation (re-encode vs. concatenate vs. latent-fuser) in terms of performance and efficiency; further fidelity analysis of the re-encoding strategy; additional comparison of two-step vs. direct sampling; and references to existing experiments, including the quantitative autoregressive vs. non-autoregressive comparison in the supplementary.
3. Clarification of contribution: Clearer statement of the main contributions and scope of our work.

We are encouraged that our detailed responses addressed the reviewers’ concerns, and that the majority of reviewers are positive about the paper. In particular, Reviewer V2su increased their score to *weak accept*, and Reviewer bEVk did not object to acceptance. We believe AutoPartGen makes a valuable contribution to the community through its demonstrated merits.

---

### Decision · Program_Chairs · 2025-09-17

**Decision:**

Accept (poster)

**Comment:**

This paper introduces AutoPartGen, an auto-regressive framework for 3D shape part generation that directly identifies and generates parts from input conditions (e.g., images or shapes) without requiring external segmentation modules. The method consists of two training stages: (1) a part autoencoder to learn compact representations of shape components, and (2) a conditioned diffusion model for autoregressive generation. During inference, the model generates 3D objects sequentially, part by part, enabling fine-grained control over structure.

The expert reviewers’ final recommendations are 3x Borderline Accept, 1x Accept, and 1x Borderline Reject, reflecting a generally positive but cautious reception after careful deliberation and consideration of the authors’ rebuttal.

The key strengths are: The method demonstrates a novel end-to-end approach for part-aware 3D generation, avoiding reliance on pre-defined or externally computed segmentations; The re-encoding strategy introduced in the rebuttal shows promising results in terms of efficiency and performance, with stable runtime and memory usage even for high-part-count models.

After a thorough discussion and a careful review of the authors' rebuttal, two primary concerns were raised:
1. **Part Order Ambiguity**:  Several reviewers questioned how the model defines its autoregressive generation order, a critical detail that was not sufficiently clarified in the original submission. While the authors address this in the rebuttal, the discussion of order ambiguity should be included in the revised version.
2. **Insufficient Ablation Study**: The initial experiments lacked the necessary ablations to isolate and validate the contributions of the paper’s core technical components. In response, the authors added experiments comparing the proposed re-encoding strategy against alternatives. These new results support the technical contribution, showing marginal improvements in performance and consistent scalability, which helps strengthen the case for the method.

While the proposed approach offers a compelling direction for structured 3D generation, the concerns around part-ordering and the depth of analysis remain significant. The authors have made meaningful progress in addressing these issues through the rebuttal, particularly by enhancing the ablation study. Therefore, despite the remaining borderline sentiment from some reviewers, the paper is recommended for acceptance. The final version should incorporate the clarifications and new experimental results presented in the rebuttal.